# Combined transcriptome and proteome profiling reveal cell-type-specific functions of *Drosophila* garland and pericardial nephrocytes
Heiko Meyer [1,2,6], Judith Bossen [3,4,6], Maren Janz [1], Xenia Müller [3], Sven Künzel [5], Thomas Roeder [3,4] ✉ & Achim Paululat [1,2] ✉

*Drosophila* nephrocytes are specialised cells that share critical functional, morphological, and molecular features with mammalian podocytes. Accordingly, nephrocytes represent a preferred invertebrate model for human glomerular disease. Here, we established a method for cell-specific isolation of the two types of *Drosophila* nephrocytes, garland and pericardial cells, from animals of different developmental stages and ages. Mass spectrometry-based proteomics and RNA-Seq-based transcriptomics were applied to characterise the proteome and transcriptome of the respective cells in an integrated and complementary manner. We observed characteristic changes in the proteome and transcriptome due to cellular ageing. Furthermore, functional enrichment analyses suggested that larval and adult nephrocytes, as well as garland and pericardial nephrocytes, fulfil distinct physiological functions. In addition, the pericardial nephrocytes were characterised by transcriptomic and proteomic profiles suggesting an atypical energy metabolism with very low oxidative phosphorylation rates. Moreover, the nephrocytes displayed typical signatures of extensive immune signalling and showed an active antimicrobial response to an infection. Factor-specific comparisons identified novel candidate proteins either expressed and secreted by the nephrocytes or sequestered by them. The data generated in this study represent a valuable basis for a more specific application of the *Drosophila* model in analysing renal cell function in health and disease.

The kidneys of vertebrates fulfil two important physiological tasks in the body. First, they produce primary and secondary urine to remove harmful nitrogen compounds that accumulate in the blood as a result of metabolic activity. Second, they reabsorb water and electrolytes while producing secondary urine, which distinguishes the kidneys as the most critical osmoregulatory organs in the vertebrate body[1]. The primary filtration of blood fluid occurs in the capillary network of the glomerulus within Bowman's capsule, and the podocytes that form the inner wall of this capsule are crucial to the filtration process. Podocytes are highly specialised cells that are connected by slit diaphragms, which are created when cell-cell adhesion proteins, such as nephrins, form a filtration apparatus with an exclusion size of approximately 70 kDa[2,3]. Components of the blood that are smaller than

70 kDa (e.g. water, ions, peptides, and specific proteins) pass through the filtration apparatus, enter the Bowman's capsule, and infiltrate the renal tubules, where they form part of the primary urine. The reabsorption of important urinary components, such as electrolytes and water, occurs in the proximal part of the renal tubules, leading to secondary urine formation[1].

Insects, including *Drosophila*, have an open circulatory system, which lacks arteries and veins; therefore, no separation of blood and body fluids occurs, and the haemolymph (a mixture of blood and body fluids) circulates freely between the organs. In addition, insects utilise two separate organ systems for detoxification, water balance, and osmoregulation, the Malpighian tubules (renal tubes) and the individual garland and pericardial nephrocytes. The combined activity of these systems is similar to that of

[1]Department of Zoology & Developmental Biology, University of Osnabrück, 49076 Osnabrück, Germany. [2]Center of Cellular Nanoanalytics Osnabrück—Cell-NanOs, 49076 Osnabrück, Germany. [3]University of Kiel, Zoology, Molecular Physiology, 24098 Kiel, Germany. [4]German Center for Lung Research (DZL), Airway Research Center North (ARCN), Kiel, Germany. [5]Max-Planck Institute for Evolutionary Biology, Plön, Germany. [6]These authors contributed equally: Heiko Meyer, Judith Bossen. ✉e-mail: troeder@zoologie.uni-kiel.de; achim.paululat@uos.de

mammalian kidneys[4]. The Malpighian tubules of *Drosophila* project both anteriorly and posteriorly into the body cavity, which is filled with hae-molymph, and active and passive transport processes along the Malpighian epithelium transfer solutes and water into the lumen of the tubes. The urine produced by the Malpighian tubules is released into the intestine and eventually becomes excreted[5]. Dysfunctional Malpighian tubules fail to excrete fluids, which causes an accumulation of excess water within the body cavity. The high physiological relevance of this impairment is reflected by the lethality that occurs in animals that lack functional septate junctions in Malpighian tubules, e.g., due to mutations in the septate junction proteins Tetraspanin-2A or Mesh[6,7].

The second critical component of the *Drosophila* renal system is a population of nephrocytes, composed of two distinct cell types: the gut-associated garland nephrocytes and the heart-associated pericardial nephrocytes. Both populations of nephrocytes have emerged as com-plementary models for renal disease in recent years and several studies have shown that an essential function of these cells is to filter toxins and proteins from the haemolymph, similar to the filtration of mammalian blood by vertebrate kidney podocytes[8–12]. *Drosophila* nephrocytes feature a sophisti-cated cell structure with elongated inward folds of the plasma membrane that merge into labyrinth channels. Haemolymph clearance, detoxification, and waste uptake are coupled with endocytosis, which occurs extensively at the membranes of the labyrinth channel system. Several molecular analyses and electron microscopy studies have confirmed this activity based on the detection of high numbers of coated vesicles in nephrocyte cortical regions[13–17]. The entrances to the labyrinth channel system are sealed by slit diaphragms, which are complex protein assemblies that consist of compo-nents similar to those found in the corresponding structures of vertebrate podocytes[4,18–21]. The slit diaphragms are critical filtration barriers that act together with the extracellular matrix (ECM) to control the influx of sub-stances from the haemolymph into the labyrinth channels. Substances that pass through these first two barriers become endocytosed and subsequently degraded within the lysosomes of the nephrocytes. In individual cases, recycling back to the haemolymph has also been observed[22].

Garland and pericardial nephrocytes are highly specialised cells that occur in low numbers in animals. Garland cells are binucleated, approxi-mately $30 \times 60\ \mu m$ in size, and found in groups of 20–25 cells in a horseshoe-like arrangement surrounding the proventriculus of the larval and adult intestine. Pericardial cells are an estimated $50 \times 100\ \mu m$ in size and represent the most voluminous cells in the organism. They have large nuclei that arise from endoreduplication[23,24]. In contrast to numerous other adult tissues and organs newly formed during pupal development, garland cells and peri-cardial nephrocytes are preserved in number and histology. Currently, there is no evidence that nephrocytes are renewed during lifetime.

Due to their functional and histological similarities to vertebrate kidney podocytes, *Drosophila* nephrocytes have become an established model for studies on clinically relevant kidney disease. Examples of nephropathies, for which the *Drosophila* model has provided fundamental insights, are numerous and growing[4,13,25–29]. To further advance the medical applicability of this model, a set of fundamental questions remains to be addressed: (1) How do cells, which are considered the most important scavenger cells in *Drosophila*, maintain their lifelong activity when they are not renewed? (2) Are garland and pericardial nephrocytes functionally equivalent or do they differ in their cellular functions? (3) Does ageing affect the general ability of the cells to remove waste material from the haemolymph?

To investigate these issues, we established a method for manually isolating garland nephrocytes (GNCs) and pericardial nephrocytes (PNCs) in a highly specific manner. Manual sorting was employed because auto-mated approaches appeared inefficient for isolating and sequencing the binucleated garland cells or polyploid pericardial cells[30]. The resultant iso-lates, collected from animals of different developmental stages and ages, consisted of highly homogeneous cell populations, and were subjected to transcriptome and proteome analyses. We found that the two types of nephrocytes rely on different types of energy metabolism and fulfil more specialised physiological functions than previously known. Moreover, both

PNCs and GNCs displayed transcriptomic signatures suggestive of an efficient immune signalling, as indicated by the enrichment of characteristic components of the Imd and Toll pathways in both types of nephrocytes.

## Results

### Isolating garland or pericardial nephrocytes from larvae and adult flies

By generating transgenic *hand*C-GFP fly lines, we established a live-cell marker that specifically labelled garland and pericardial nephrocytes without affecting their differentiation and function[31]. The *hand*C-GFP line was homozygous viable, showed regular fitness and wild-type longevity, and was successfully used in previous studies[32–34]. Herein, we established a method for isolating individual cells from larvae or adult animals of different ages. PNCs are located close to the heart tube, to which they are linked by an extensive cardiac extracellular matrix (ECM) network[14]; therefore, collagenase treat-ment was required to release the cells from the ECM meshwork. Subse-quently, the pericardial nephrocytes, which were easily recognisable by their *hand*C-GFP fluorescence, were collected using a microinjector device (Fig. 1). GNCs were isolated similarly after exposing the oesophagus and proventriculus with the attached cells to the collagenase solution (see 'Materials and methods' section for details). The quality and homogeneity of the individual preparations were confirmed by microscopic analyses (Fig. 1). Based on this method, cell-specific transcriptome (GSE250029, Gene Expression Omnibus, https://www.ncbi.nlm.nih.gov/geo) and proteome analyses (PXD047262, ProteomeXchange Consortium, https://www.proteomexchange.org) of the two nephrocyte types were performed. The transcriptome analysis required amplification of the transcripts, as recently established for adult tracheal analyses in *Drosophila*[35]. The number of iso-lated cells used for transcriptome or proteome analysis is listed in Table 1.

### Canonical functions of adult nephrocytes are reflected in their specific transcriptome and proteome profiles

To determine the core characteristics of the nephrocytes, the transcriptome and proteome profiles of PNCs and GNCs isolated from 1-week-old females were compared with the corresponding profiles of age-matched complete animals (Fig. 2). Relative to complete animals, the PNCs and GNCs exhibited distinct differences in gene expression, as visualised by a heatmap (Fig. 2A). In total, 6279 (PNCs) and 5551 (GNCs) differentially expressed genes (DEGs) were detected. Both cell types displayed several upregulated genes (PNCs: 2765, GNCs: 2323; Fig. 2C). Of note, the transcripts encoding the nephrocyte-specific receptors Amnionless and Cubilin (Cubn), along with Cubilin 2 (Cubn2), which are essential for protein reabsorption in nephrocytes[36], were highly enriched in the transcriptomes of both nephrocyte types, with fold changes ranging from 145 (GNCs, *Cubn*) to 974 (PNCs, *Amnionless*; Fig. 2B). This was also valid for the *Kruppel-like factor 15* (*Klf15*), *sticks and stones* (*sns*) and *kin of irre* (*kirre*), all of which are essential for nephrocyte function (Fig. 2B). These results indicate a high specificity of the applied cell isolation protocol. A comparison of the upregulated genes in both PNCs and GNCs identified 1438 common genes (Fig. 2C). Gene ontology (GO) and Kyoto Encyclopedia of Genes and Genomes (KEGG) pathway enrichment analyses were performed using these genes as a basis to determine the canonical functions shared by the two cell types. We identified enrichments in endocytosis, apoptosis, autophagy, mitophagy and lysosomal function (Fig. 2D) and abundancies in numerous signalling pathways, including FoxO, Hippo, MAPK, Toll, Imd and the longevity-regulating pathway. Of note, no significant apoptosis has been reported for the *Drosophila* nephrocytes. Thus, the occurrence of this GO term should be viewed with caution. The GO annotation may rather be based on the fact that components of the MAPK, Hippo, Toll, and Imd signalling pathways were enriched in the nephrocytes (Fig. 2D). The induction of apoptosis is one of the multiple cellular functions of these pathways. However, the corresponding signalling is likely not involved in promoting apoptosis in nephrocytes. A more detailed analysis of the genes associated with the longevity-regulating pathway (Fig. 2E) revealed enrichment of the two known longevity genes *Sirt1* and *foxo*. In addition to

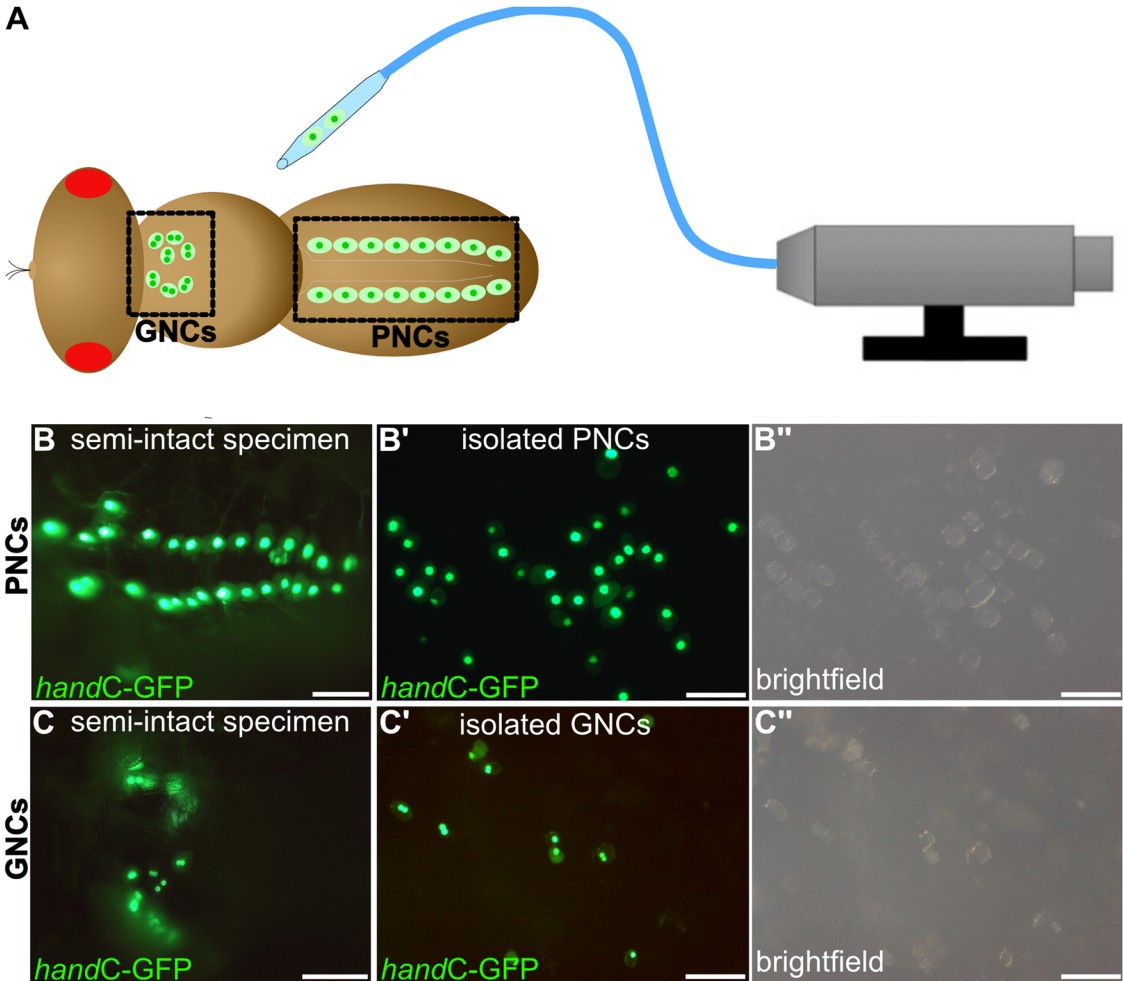

**Fig. 1 | Experimental setup for isolating garland or pericardial nephrocytes from adult flies. A** *hand*C-GFP expressing female flies were immobilised, ventral side up, and either garland (GNCs) or pericardial nephrocytes (PNCs) were isolated using a CellTram equipped with a thin glass capillary. Semi-intact specimen before collagenase treatment with GFP-labelled pericardial cells (**B**, one nucleus per cell) or garland cells (**C**, two nuclei per cell). (B', B", C', C") After collagenase treatment. The dissociated nephrocytes are isolated and collected. Scale bars: 100 μm.

*foxo*, several genes related to the insulin signalling pathway were enriched (*Ilp6*, *InR*, *Thor*, *Pi3K92E*, *chico*). Moreover, we recognised several heat-shock protein genes that exhibited increased expression in both nephrocyte types relative to complete animals.

Similar to the transcriptome data, characteristic differences between 1-week-old PNCs and complete animals were also observed at the protein level

**Table 1 | Number of utilised cells/animals per measurement**

| Tissue and developmental stage | Transcriptomics | Proteomics |
|---|---|---|
| PNCs, 3rd instar larvae | 100 cells | 200 cells |
| GNCs, 3rd instar larvae | 30 cells | 50 cells |
| PNCs, 1-week adults | 100 cells | 200 cells |
| GNCs, 1-week adults | 30 cells | 50 cells |
| PNCs, 3-week adults | 100 cells | 200 cells |
| GNCs, 5-week adults | n.d. | 50 cells |
| PNCs, 5-week adults | 100 cells | 200 cells |
| 1-week-old flies (total) | 5 animals | 5 animals |

For transcriptome/proteome analyses, each measurement was repeated at least four times using individually isolated cells.
*n.d.* not determined.

(Fig. 2F and Supplementary Data 1A). We found proteomic signatures, which indicated that the pathways associated with the cell scavenger function were highly enriched in the PNCs and factors specific to efficient glycan degradation were abundant (e.g. Hexo1, Hexo2, and Ect3). Furthermore, we observed increased levels of proteins that mediate endocytosis and vesicle maturation (e.g. Rab7, Rab11, Snx1, Vps35, and Ist1), autophagy (e.g. Rab1, Rab7, and Atg3) and lysosomal function (e.g. Acph-1, Sap-r, Hexo2, and Ect3; Fig. 2G and Supplementary Data 1B). Finally, Amnionless and Cubilin were abundantly detected in the PNCs (Supplementary Data 1A). Thus, the obtained transcriptome and proteome profiles distinctly reflected the predominant physiological functionality of the pericardial nephrocytes, thereby supporting the overall quality and applicability of our experimental setup. In addition to the expected characteristics, the PNC protein profiles indicated comparatively low aerobic respiration rates and protein biosynthesis levels compared to complete animals. While the former effect was mainly related to the reduced abundance of factors required for oxidative phosphorylation (e.g. Cyt-c1, SdhC, and ATPsynG) and TCA cycle progression (e.g. Mdh1, Mdh2, and Fum1), decreased protein biosynthesis rates were indicated by reduced amounts of various ribosomal proteins (e.g. RpLP0, RpLP1, and RpS2). Finally, the PNC-specific protein profiles showed a general decrease in the metabolic activity of these cells, particularly concerning amino acid and fatty acid metabolism (Fig. 2H and Supplementary Data 1C). The latter observation may indicate that at least part of the endocytosed cargo is stored within the cells, rather than metabolised.

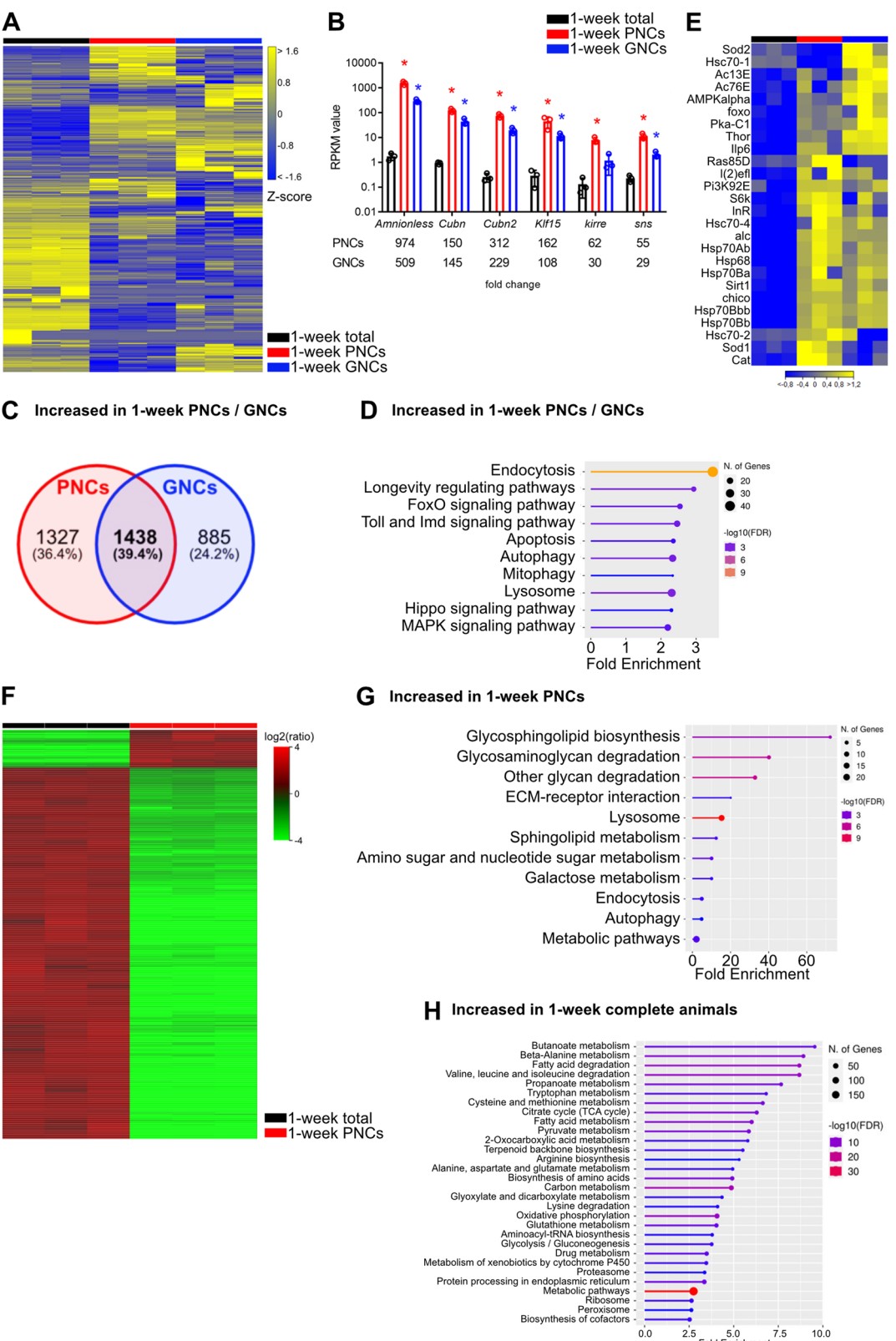

**Fig. 2 | Core characteristics of nephrocytes at the transcript and protein level.**
**A** Heatmap of RNAseq replicates (minimum fold change 1.5, FDR *p* value cutoff
0.05) from total animals (black bar), PNCs (red bar) and GNCs (blue bar). **B** RPKM
values of *Amnionless*, Cubilin (*Cubn*) and Cubilin 2 (*Cubn2*), Kruppel-like factor 15
(*Klf15*), *sticks and stones* (*sns*), and *kin of irre* (*kirre*) in the three groups. The
corresponding fold changes relative to complete animals are shown in the lower
panel. Asterisks indicate a differential expression of the respective gene compared to
the total animals. **C** Venn diagram of 2765 upregulated genes in PNCs and 2323

upregulated genes in GNCs, relative to total animals; 1438 genes (39.4%) were shared
between the two groups. **D** KEGG pathway enrichment analysis of the shared genes.
**E** Heatmap of genes associated with the 'Longevity regulating pathway' (KEGG)
depicting alterations for the RNA-seq groups and replicates. **F** Heatmap of proteome
replicates from total animals (black bar) and PNCs (red bar). **G** KEGG pathways
enriched in PNCs, relative to complete animals. **H** KEGG pathways enriched in
complete animals, relative to PNCs.

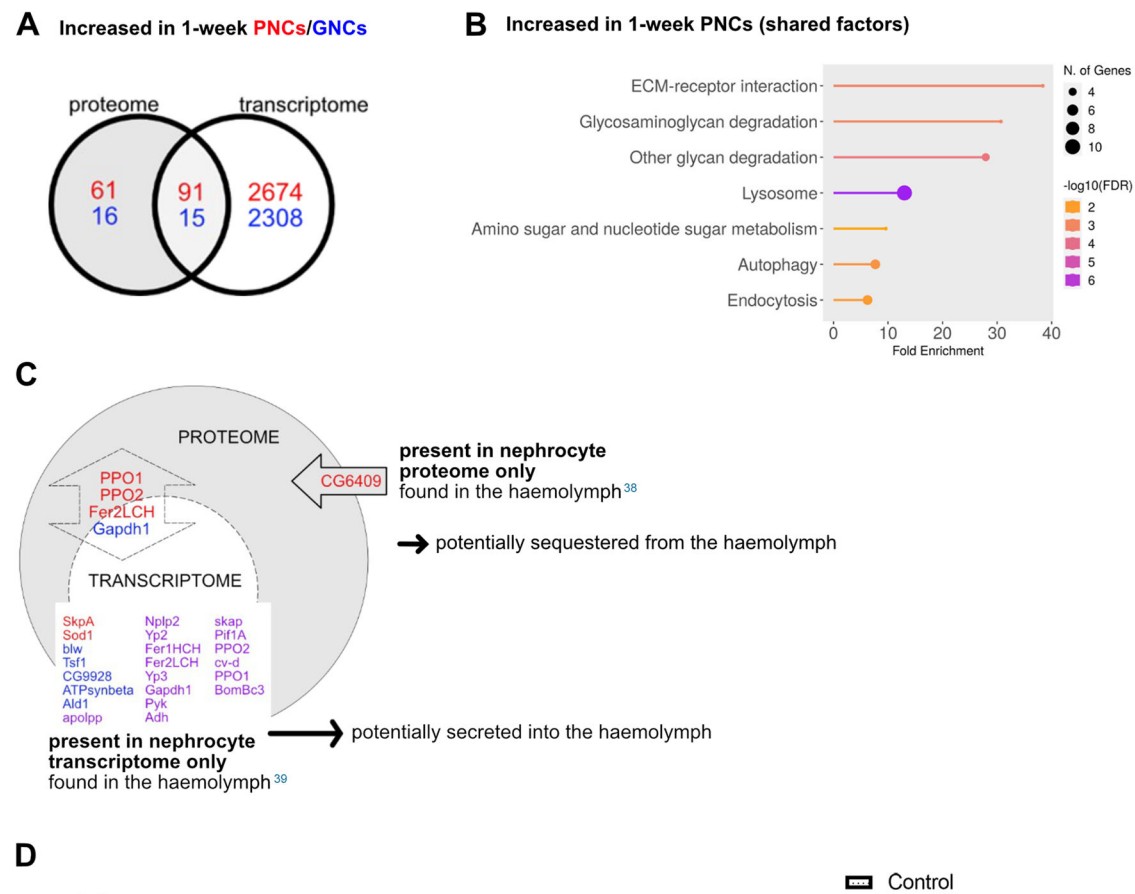

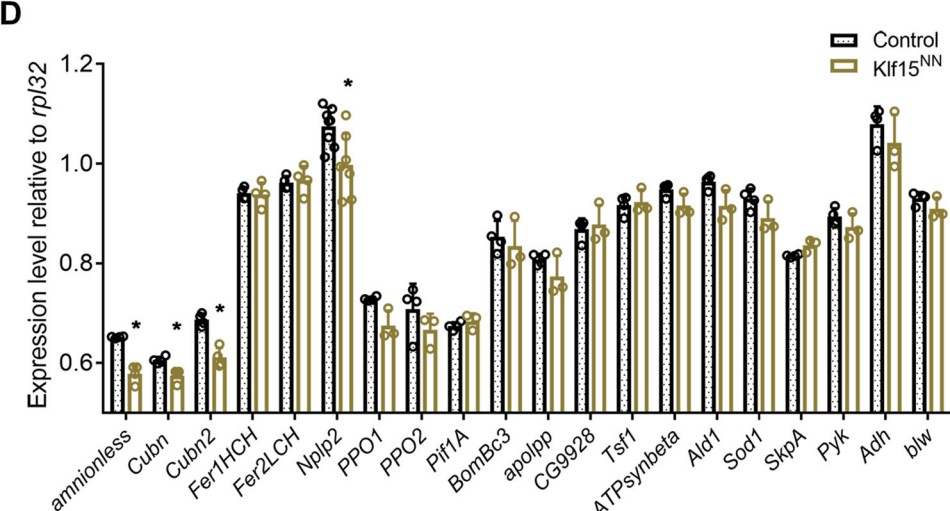

**Fig. 3 | Correlative analyses of the nephrocyte transcriptome and proteome.**
**A** Red: Venn diagram of 152 proteins enriched in PNCs and 2765 genes upregulated in PNCs, relative to total animals; 91 proteins/genes were shared between the two groups, which corresponds to 59.9% of the enriched proteins. Blue: Venn diagram of 31 proteins enriched in GNCs and 2323 genes upregulated in GNCs, relative to total animals; 15 proteins/genes were shared between the two groups, which corresponds to 48.4% of the enriched proteins. **B** KEGG pathway enrichment analysis of the

shared factors. **C** Analysis of factors present only in the nephrocyte proteome (arrow), only in the nephrocyte transcriptome (dashed circle), or in both (double arrow). Factors in red font were detected only in PNCs, factors in blue font only in GNCs, and factors in purple font were detected in both cell types. **D** Quantitative RT-PCR of the genes expressed in 1-week-old animals with depleted nephrocytes (*Klf15^NN*), relative to wildtype (*w^1118*). Mean and SD are shown, $n = 4$, *$p = <0.05$, Mann–Whitney $U$ test.

## Correlative analyses of the nephrocyte transcriptome and proteome can identify factors that are sequestered or secreted by the cells

In addition to analysing the transcriptomic and proteomic datasets individually, we also evaluated the share of overlapping proteins/genes that exhibited significantly altered amounts/expression levels in the nephrocytes, compared to complete animals. Increased transcript levels were detected for about 60% (PNCs) or 48% (GNCs) of the proteins that exhibited increased

abundance in the PNCs (91 of 152) or GNCs (15 of 31), respectively (Fig. 3A). A KEGG enrichment analysis of these overlapping proteins/genes in the PNCs identified elevated ECM-receptor interaction, glycan degradation, lysosomal function, autophagy and endocytosis as major pathways (Fig. 3B). Despite this considerable overlap between the proteomic and transcriptomic data, several proteins with altered abundance had no correspondingly affected transcript. Because nephrocytes are involved in regulating the amount of secreted proteins that circulate in the haemolymph[37],

we analysed a set of proteomic studies that focused on haemolymph composition[38,39], and cross-correlated the identified proteins with our data (Fig. 3C). As a result, we found a single haemolymph circulating protein that was also present in our proteomic dataset but absent in the transcriptome data. This yet uncharacterised protein (CG6409, Fig. 3C) is likely sequestered by the PNCs, rather than produced by the cells. In addition, we identified a set of proteins present in the previously characterised haemolymph proteome and in our current proteome and transcriptome datasets. These factors (e.g. PPO1, PPO2, Fer2LCH, Fig. 3C) are evidently expressed by the nephrocytes, but presumably also sequestered by them.

On the other hand, we also identified several haemolymph circulating factors present in the transcriptome of both PNCs and GNCs, but absent in at least one of the corresponding proteomic datasets (Fig. 3C, purple). These factors may represent gene products expressed by the nephrocytes and secreted into the haemolymph but not efficiently sequestered. To evaluate whether the nephrocytes represent the main source of these factors, we compared the nephrocyte expression levels of some of the corresponding genes to the respective levels in other tissues. Here, we applied animals lacking nephrocytes (Klf15[NN][20]) and performed qRT-PCR analyses relative to animals with unaffected nephrocytes. In addition to the haemolymph circulating factors, we included the nephrocyte-specific genes Amnionless, Cubilin (Cubn) and Cubilin 2 (Cubn2) in our analysis. All three genes exhibited a reduced expression in Klf15[NN] animals, compared to control animals, which confirms their largely specific expression in nephrocytes. Regarding the selected haemolymph circulating factors, only Nplp2 and PPO1 showed a significantly reduced expression in Klf15[NN] animals, compared to control animals with unaffected nephrocytes (Fig. 3D). Thus, at least for these two factors, the nephrocytes appear to be a significant site of production.

### PNCs and GNCs display transcriptomic signatures indicative of elevated immune signalling

In addition to the characteristics described above, enrichment of the 'Toll and Imd signalling' KEGG pathway was observed in both the 1-week-old PNC and GNC transcriptome, compared to complete age-matched animals (Fig. 2D). To support the indication of elevated immune signalling in these cells, we selected six antimicrobial peptides (AMPs) as target genes of the two pathways and visualised transcription in the PNCs via GFP-reporter expression controlled by the individual gene-specific promotors[40] (Fig. 4A). Most of the constructs showed clear signals in the PNCs, except for Dpt that exhibited a reduced signal intensity. In addition, we analysed the fold changes of the associated AMPs in our transcriptome dataset. We found that 16 AMPs were expressed in the PNCs, including various Bomanins and immune-induced peptides (IMs). Among these, IM18, AttD, Mtk, DptB and CecA2 exhibited ≥10-fold higher abundances relative to complete animals (Fig. 4B). For the GNCs, Mtk was the only AMP that was strongly expressed (fold change 57.5, Fig. 4B). We confirmed the expression of Mtk in both GNCs and PNCs using the Mtk-GFP reporter line (Fig. 4D). Furthermore, the components of the Toll and Imd pathways and the corresponding components of the tightly connected JNK pathway were analysed in further detail using the individual fold change data of the comparisons to complete animals as a basis to evaluate the differences between the cell types (Fig. 4C, PNCs vs. total (red fill), GNCs vs. total (blue border)). Most of the Toll pathway components, which recognises fungi and gram-positive bacteria, were enriched in the PNCs, while only three of these were enriched in the GNCs. In contrast, most of the components of the Imd pathway were strongly expressed in both nephrocyte types. In particular, PGRP-LA, PGRP-LF, pirk, Relish, and Ankyrin were highly enriched in PNCs, indicating a strong upregulation of the Imd pathway in these cells. Toll and Imd signalling form part of the transcriptional regulation of distinct defence response genes (Fig. 4B, C). The JNK pathway is tightly connected to the Imd pathway via Tak1, and the pathway ligand Eiger was strongly expressed in PNCs and GNCs, relative to complete animals, while the receptor and intracellular components dTraf2, Tak1, and hep were not detected in both cell types. Beginning with the Bsk level, all further pathway components

were expressed, especially the JNK transcription factors Jra and Kay (AP1), with Ets21c showing a particularly strong expression in both nephrocyte types. Regarding the basic JNK transcription factors, Jra and Kay, the most prominent target genes dpp, puc, and Mmp1 were expressed in both PNCs and GNCs (Fig. 4E). Given that the transcription factor Ets21c is associated with JNK signalling[41,42], we analysed known target genes of this protein[42] and found that among the 22 possible factors, 16 were expressed in the nephrocytes (Fig. 4F). Overall, these data indicate strong activities of the respective immune signalling pathways in the PNCs and GNCs and suggest an essential function of the cells in modulating general immune responses. This function may go beyond the recently noted ability of the cells to remove microbiota-derived pathogen-associated molecular patterns (PAMPs) from the circulation via endocytosis and perform subsequent lysosomal degradation[43]. To further substantiate these indications and thus confirm the biological relevance of the transcriptomic signatures, we performed systemic bacterial infection experiments using Pectobacterium carotovorum (gram-negative bacteria, former name Erwinia carotovora carotovora) as a pathogen and Drosomycin-GFP (Drs-GFP) and Defensin-GFP (Def-GFP) reporter lines as readouts for increased expression of antimicrobial peptides[40]. We found that expression of the Drs-GFP reporter was strongly increased in PNCs in response to infection, compared to uninfected and sterile-pricked control animals (Fig. 4G, G', G", I). The same effect was observed in fat body cells, confirming that both cell types exhibit significant immune competence (Fig. 4J). On the other hand, expression of the gene encoding Defensin, an antimicrobial peptide active predominantly against gram-positive bacteria[40], was not induced, which corroborates the specificity of the immune response (Fig. 4H, H', H", K, L).

### The transcriptome and proteome profiles of garland and pericardial nephrocytes indicate distinct functional and metabolic differences between the cell types

Both PNCs and GNCs are classified as nephrocytes, predominantly based on their ultrastructure, which consists of characteristic membrane invaginations that induce the formation of a surface-enlarged labyrinth channel system to optimise endocytosis efficiency. In addition, both cell types possess slit diaphragms and efficient endocytic mechanisms for the uptake of various compounds from the haemolymph[4]. To determine whether GNCs and PNCs indeed have equivalent physiologies and functionalities, we analysed and compared their respective transcriptomes and proteomes and detected several interesting differences between the two cell types.

The transcriptome comparison revealed 920 upregulated genes in GNCs, relative to PNCs, and 1723 upregulated genes in PNCs, relative to GNCs (Fig. 5A). The predominant profiles were indicative of an increased reactive oxygen species metabolism (peroxisome), as well as augmented protein degradative abilities (lysosome and proteasome) and elevated endocytosis efficiencies in PNCs, compared to age-matched GNCs (Fig. 5B). In addition, the cellular energy metabolisms appeared to be different between the two cell types. The GNCs showed specific enrichments in the 'Oxidative phosphorylation', 'Glycolysis/Gluconeogenesis' and 'Citrate cycle (TCA cycle)' metabolic pathways, relative to PNCs (Fig. 6A). The corresponding proteome analyses largely supported the transcriptome data (Fig. 5A' and Supplementary Data 2A). The proteome profiles of the PNCs again indicated a significantly elevated endocytosis efficiency and an increased capacity for glycan and protein degradation when compared to aged-matched GNCs (Fig. 5B'). The underlying proteins with increased abundances included Gal, Hexo1, Rab5, Rab7, Rab11, Snx3, Snx6, and Vps35 (Supplementary Data 2B). Moreover, in line with the transcriptome data, the GNC proteome profiles indicated substantially augmented oxidative phosphorylation and TCA-cycle rates relative to PNCs (Fig. 6A'). This apparent increase in oxidative phosphorylation mainly resulted from the enhanced levels of proteins involved in mitochondrial electron transport, with subunits of all respiratory chain complexes detected (e.g. RFeSP, UQCR-14, Cyt-c1, SdhC, ND-23, ATPsynG, and ATPsynδ; Fig. 6B), whereas the higher TCA-cycle rates were indicated by elevated levels of enzymes essential to

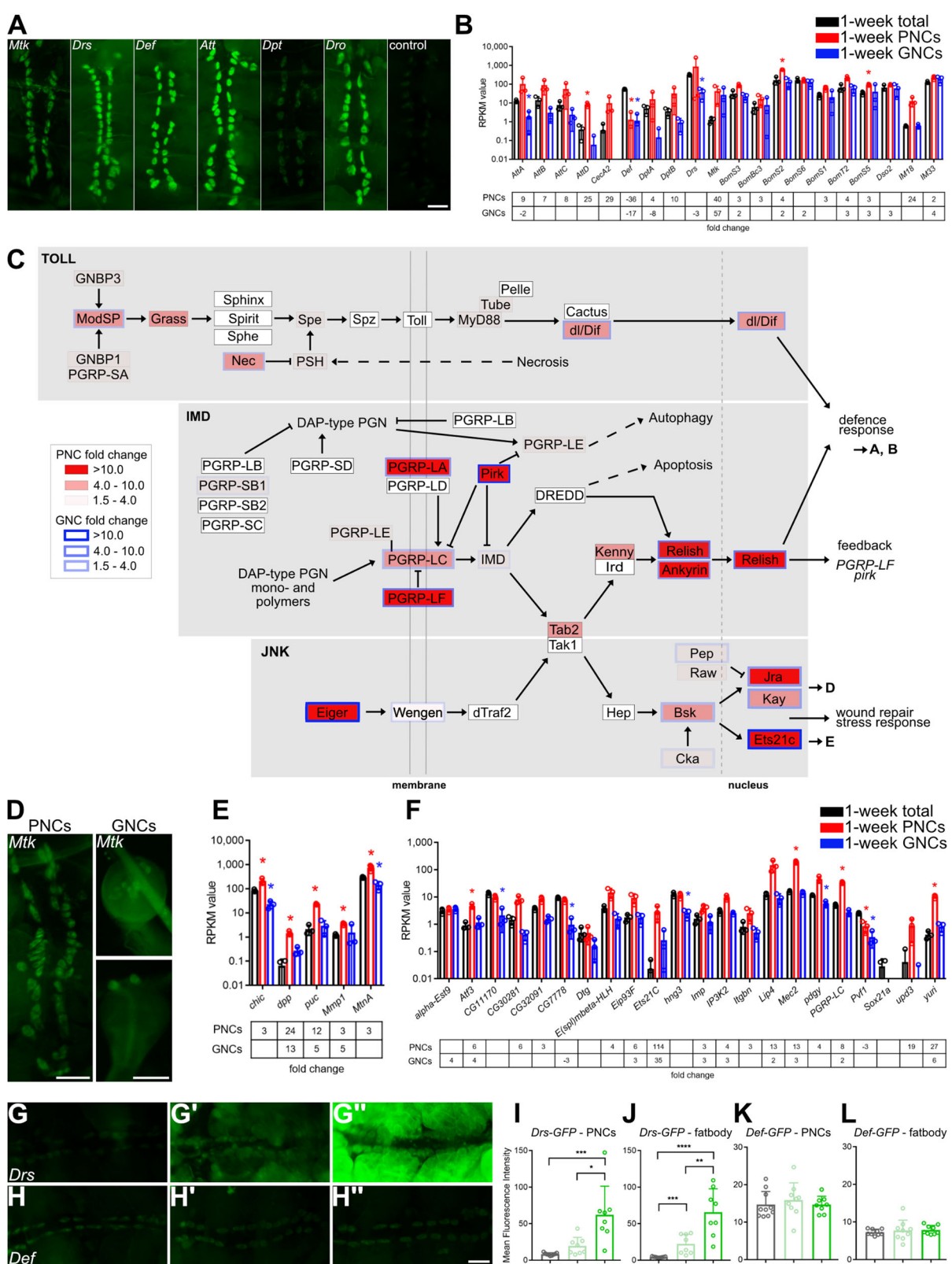

TCA-cycle progression (e.g. ScsβA, Idh3b, Pdhb, Mdh2, and Fum1; Fig. 6A′ and Supplementary Data 2C).

We noted an analogous pattern for the 1-week-old GNCs compared to age-matched complete animals. The corresponding proteomic signatures similarly indicated the occurrence of augmented oxidative phosphorylation in the GNCs. However, no signatures relating to

increased efficiencies in lysosomal function or glycan degradation were evident relative to the complete animals (Fig. S1A′–C′ and Supplementary Data 3A–C). The latter aspect indicates that in adult Drosophila, scavenging and disposal of haemolymph circulating glycans or proteoglycans is performed more efficiently by the PNCs than by the GNCs.

**Fig. 4 | Nephrocytes exhibit increased expression of Toll, Imd and JNK signalling components. A** Expression of different antimicrobial peptides in PNCs indicated by GFP reporter activity. Control animals ($w^{1118}$) do not exhibit any signal above background. Scale bar: 100 μm. **B** RNAseq RPKM (Reads Per Kilobase Million) values of defence response target genes in PNCs and GNCs, relative to complete animals (total). The lower panel depicts the expression fold changes in PNCs and GNCs relative to the total animals. **C** Schematic representation of Toll, Imd and JNK signalling pathways. RNAseq fold change of PNCs (red background) and GNCs (blue frame), relative to total animals, is indicated by colour coding. Insignificant differential expression of a component is indicated by a white fill (PNCs) or a fine black border (GNCs). Connected pathways are depicted by dashed arrows. Targeted processes downstream of the three pathways are shown to the right. **D** *Mtk-GFP*

reporter expression in PNCs (left) and GNCs (right). Scale bars: 100 μm. **E** RPKM values of JNK target genes regulated by Jra/Kay in PNCs (red) and GNCs (blue), relative to complete animals. The lower panel depicts the corresponding expression fold changes. **F** RPKM values of JNK target genes regulated by Ets21c in PNCs (red) and GNCs (blue), relative to complete animals. The lower panel depicts the corresponding expression fold changes. Asterisks indicate a differential expression of the respective gene compared to the total animals (**B**, **E**, **F**). **G–L** Uninfected PNCs of *Drs-GFP* and *Def-GFP* reporter lines (**G**, **H**) compared to sterile infected (G', H') and *Pectobacterium carotovorum* (*P.c.*) infected adults (G", H"). Scale bar: 1 mm. The mean fluorescence intensity of PNCs (**I**, **K**) and the adjacent fatbody cells (**J**, **L**) was quantified. Mean and SD are shown, $n > 8$, $*p = <0.05$, $**p = <0.01$, $***p = <0.001$, $****p = <0.0001$, Mann–Whitney $U$ test.

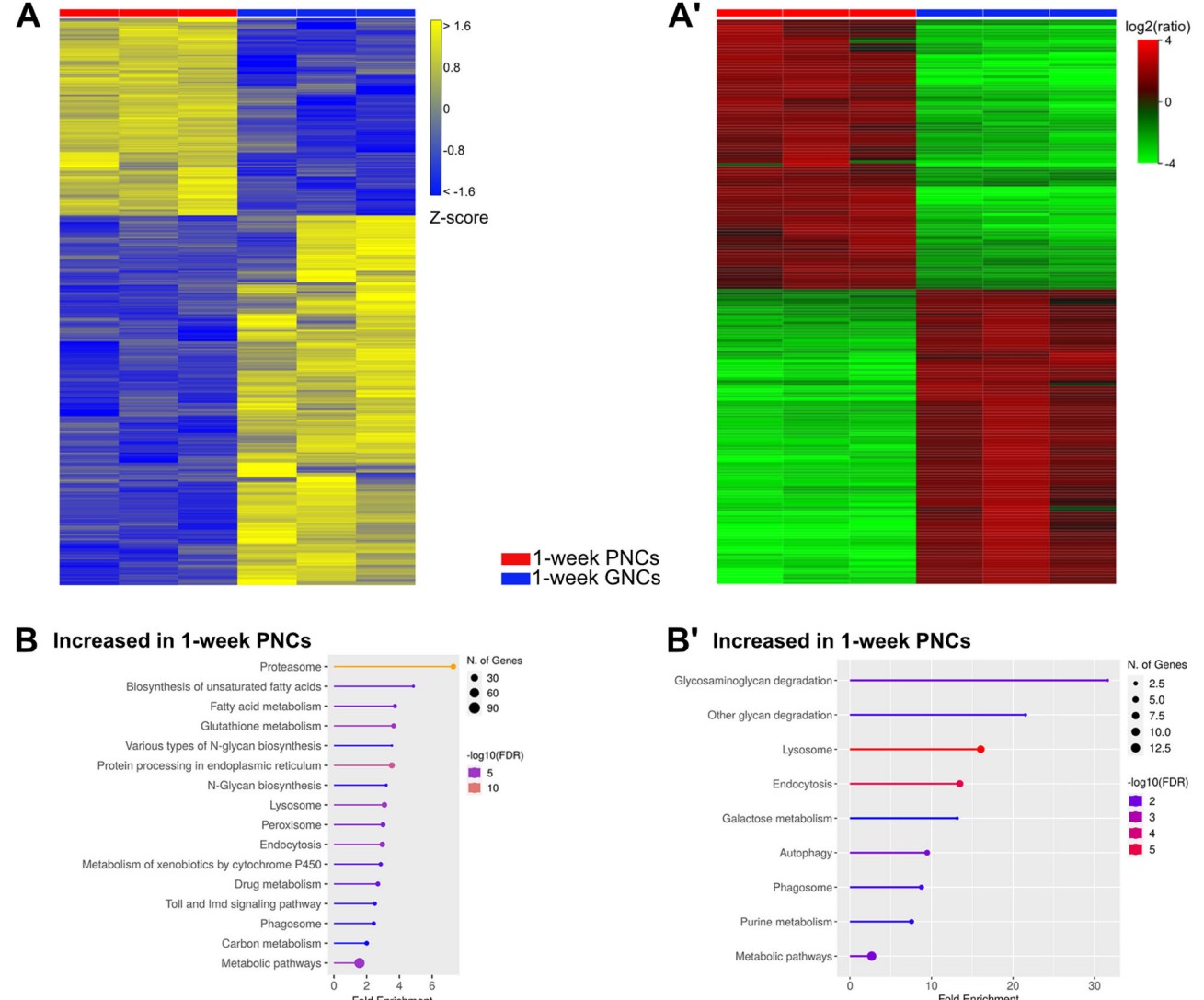

**Fig. 5 | Nephrocytes exhibit cell-type-specific transcriptomic and proteomic signatures. A** Heatmap of RNAseq replicates (minimum fold change 1.5, FDR *p* value cutoff 0.05) from 1-week-old PNCs (red bar) and 1-week-old GNCs (blue bar). (A´) Heatmap of proteome replicates from 1-week-old PNCs (red bar) and 1-

week-old GNCs (blue bar). **B** KEGG pathways (transcriptome-based) enriched in 1-week-old PNCs, relative to 1-week-old GNCs. (B´) KEGG pathways (proteome-based) enriched in 1-week-old PNCs, relative to 1-week-old GNCs.

The 'Oxidative phosphorylation' KEGG pathway exhibited a highly significant enrichment in GNCs, relative to both PNCs and complete animals (transcriptome- and proteome-based, Figs. 6A, A´ and S1). Therefore, we analysed the factors underlying this pathway (https://www.kegg.jp/pathway/map=dme00190) in more detail (Fig. 6B; transcriptome = purple background, proteome = green frame). All five respiratory chain complexes were represented in our data. Most of the Complex I subunits (NADH

dehydrogenase) were significantly enriched in the transcriptome and largely overlapped with the proteome data, except for subunits ND1 to ND6. Similarly, all subunits of Complex II (Succinate dehydrogenase) were affected, at least in terms of the transcriptome, and considerable overlap was also evident between the transcriptome and proteome data of the Complex III (Cytochrome c reductase) and Complex IV (Cytochrome c oxidase) subunits. Furthermore, the subunits of the F-type ATPase were strongly

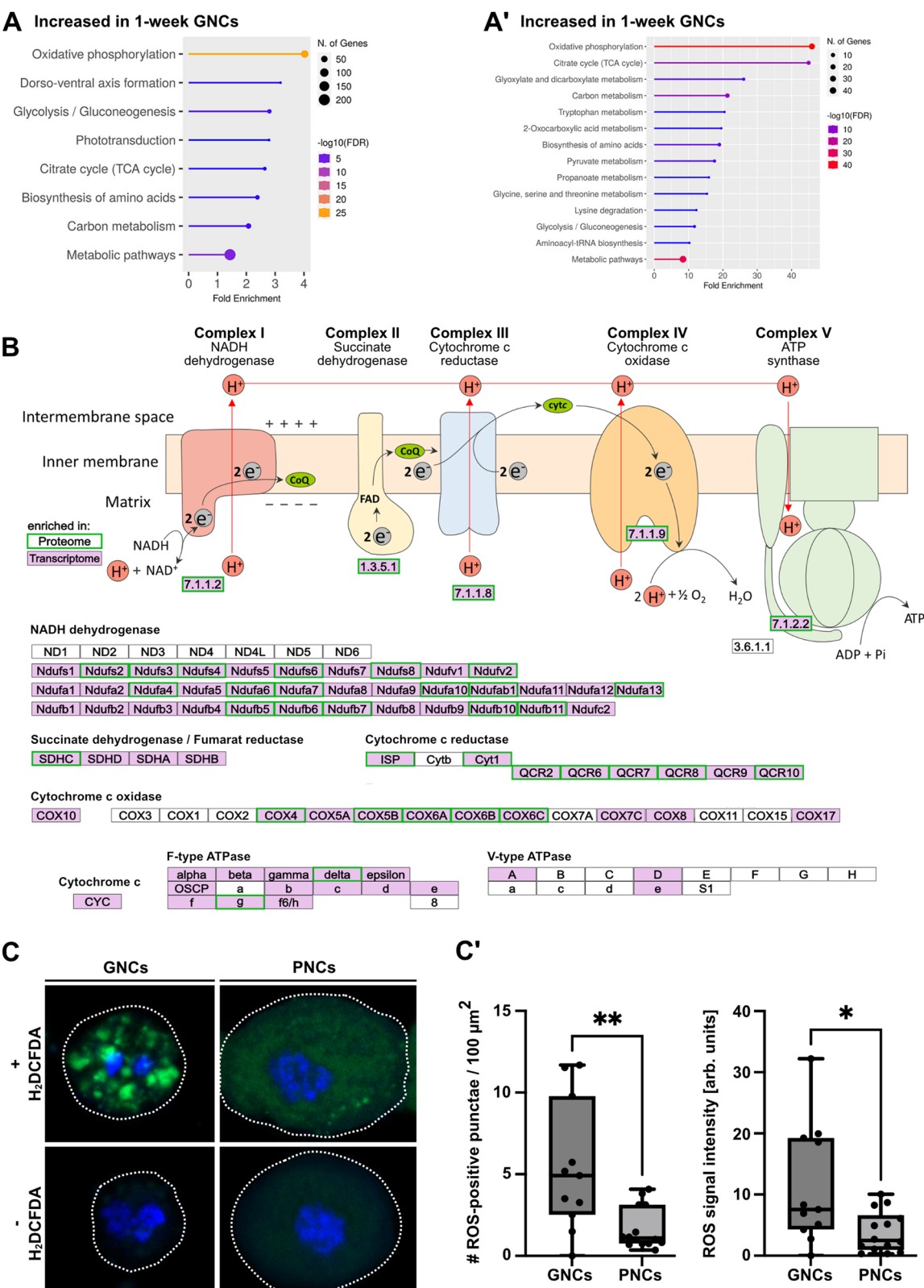

represented. In contrast, for the V-ATPase only subunits A, D, and e showed increased expression in GNCs when compared to PNCs (Fig. 6B). To verify the indication of a substantially increased oxidative phosphorylation rate in GNCs, relative to PNCs, we analysed the formation of reactive oxygen species (ROS) in the garland and pericardial nephrocytes. ROS are by-products of oxidative phosphorylation; therefore, measuring ROS

production in a cell-specific manner allows for estimating oxidative phosphorylation rates in the respective cells or tissues[44–46]. The overall ROS signal intensity and the number of ROS-positive punctae per cell were significantly increased in GNCs, compared to PNCs (Fig. 6C, C´). In addition, Mito-Tracker stainings revealed a slightly but significantly increased abundance of mitochondria in GNCs, relative to PNCs (Fig. S3). These results further

**Fig. 6 | Nephrocytes exhibit transcriptomic and proteomic signatures indicative of a cell-type-specific energy metabolism. A** KEGG pathways (transcriptome-based) enriched in 1-week-old GNCs, relative to 1-week-old PNCs. (**A´**) KEGG pathways (proteome-based) enriched in 1-week-old GNCs, relative to 1-week-old PNCs. **B** Scheme depicting factors relevant to oxidative phosphorylation (KEGG pathway). Transcripts (purple background) and proteins (green frame) with significantly increased abundance in GNCs, relative to PNCs, are highlighted. **C** Isolated GNCs and PNCs with or without the addition of a ROS indicator (H$_2$DCFDA, green channel). DAPI was used as a counter stain to visualise the nuclei (blue channel). A dotted line indicates the cell perimeter. Scale bars: 10 μm. (**C´**) The number (#) of ROS-positive punctae and the overall ROS signal intensity were determined by pixel intensity measurements. Signal intensities were normalised to corresponding control cells without the ROS indicator. For both boxplots, the centre line indicates the median, the upper and lower bounds indicate the 75th and 25th percentiles, respectively, and the whiskers indicate the minimum and maximum. For each analysis, a minimum of 10 individual cells was measured. Asterisks indicate statistically significant differences between the two cell types (*$p < 0.05$, **$p < 0.01$, Mann–Whitney $U$ test).

support an increased oxidative phosphorylation rate in GNCs compared to PNCs, which was already indicated by the analysis of our transcriptome / proteome data (Fig. 6A, A´, B).

### The transcriptome and proteome profiles of pericardial and garland nephrocytes exhibit characteristic changes during ageing

A previous study demonstrated that nephrocytes undergo structural degeneration with ageing[16]. To investigate the underlying molecular changes in detail, we compared the transcriptome and proteome of 1-week-old, 3-week-old, and 5-week-old PNCs. In addition, by including larval PNCs in the analysis, we assessed the changes during development. Here, the extent to which the transcriptome and proteome of nephrocytes adapt during the larval-to-adult transition was analysed.

At the transcriptional level, the larval PNCs possessed 1390 upregulated and 1388 downregulated genes, relative to PNCs from 1-week-old adult animals (Fig. 7A). KEGG pathway analyses revealed that the genes with higher expression in larvae included factors related to DNA replication and the associated repair processes. 'Lysine degradation' was the fourth KEGG pathway with enrichment in larval PNCs, compared to adult PNCs (Fig. 7B). Evaluation of the pathways that were enriched in the larval PNC proteome (fatty acid degradation, ribosome, N-glycan biosynthesis, protein processing in endoplasmic reticulum, and glycolysis/gluconeogenesis, Fig. 7A', B´ and Supplementary Data 4A, B) indicated a substantial protein production capacity, which likely included a considerable share of secreted proteins that require processing within the ER.

For the GNCs, the larval cells had 1280 upregulated and 1396 downregulated genes, relative to GNCs from 1-week-old adults (Fig. S4A). KEGG pathway analyses revealed that the larval cells were enriched in degradative processes, e.g. proteasome, fatty acid degradation, and lysosome. On the other hand, biosynthesis processes were identified including insect hormone biosynthesis, N-glycan biosynthesis, protein processing in endoplasmic reticulum, and protein export (Fig. S4B). Similar to the results for the larval PNCs, the larval GNC proteome showed little overlap with the corresponding transcriptome in terms of identified KEGG pathways. Only fatty acid degradation was identified as a significantly enriched process in both datasets (Fig. S4B, B´). Whether this phenomenon was due to reduced transcript or protein stability in the larval nephrocytes or to other molecular events such as altered translation efficiency remained unclear. Further characterisation of the larval GNC proteome revealed that the cells were characterised by a number of KEGG pathways that were also enriched in the proteome of larval PNCs, including ribosome, fatty acid degradation, and glycolysis/gluconeogenesis. On the other hand, certain pathways were exclusive to the GNCs (ECM-receptor interaction; beta-alanine metabolism; valine, leucine, and isoleucine degradation; glutathione metabolism, Fig. S4B´).

Vice versa, numerous metabolic pathways appeared to be underrepresented in larval PNCs and GNCs, relative to their adult counterparts. In the transcriptome, 1-week-old PNCs exhibited enrichments in KEGG pathways associated with fatty acid, amino acid (in particular tyrosine, valine, leucine, and isoleucine) and carbohydrate metabolism (Fig. 7C). Moreover, the appearance of the 'Toll and Imd signalling' KEGG pathway in 1-week-old PNCs and the reduced expression of several peptidoglycan recognition receptors (PGRPs: *PGRP-SA*, *PGRP-LC*, and *PGRP-LF*),

downstream immune signalling pathway components (*modSP*, *SPE*, *psh*, *nec*, *pirk*, *Tab2*, *Jra*, *Ank2*, *Rel*, and *p38a*), and target genes (*CecA1*, *BomS2*, *IM4*, *BomS3*, and *Drsl4*) of the Toll and Imd pathways in larval PNCs implied that the immune response is more relevant in adult PNCs than in the larval cells (Fig. 7C).

Similar to the transcriptome data, the analyses of the adult PNC proteome indicated elevated levels of specific metabolic pathways, particularly those of the amino acid and carbohydrate metabolisms, relative to 3rd instar larval PNCs (Fig. 7C'). In addition, the proteome profile of the larval PNCs indicated a reduced ability to degrade glycans (reduced levels of Ect3 and CG18278) and a decreased abundance of lysosomal acid hydrolases (Acph-1, Sap-r), relative to adult PNCs (Fig. 7C´ and Supplementary Data 4C). These results suggest that the pericardial nephrocytes attain their full ability to scavenge and degrade circulating waste substances from the haemolymph only in the adult stage. Interestingly, tracer studies measuring the endocytic uptake of Cy3-labelled avidin confirmed an age-dependent decline in uptake efficiency for adult PNCs (1-week-old vs. 5-week-old cells). In larval PNCs, however, we observed an uptake efficiency that was even higher than the uptake rate of 1-week-old adult cells (Fig. S5). Thus, while the larval PNCs appear to be rather active scavengers, degradation of the sequestered material likely occurs predominantly in adult cells.

For the 1-week-old GNCs, transcriptome analyses predominantly identified KEGG pathways related to the cellular energy metabolism. In particular, oxidative phosphorylation, glycolysis/gluconeogenesis, TCA cycle, pyruvate metabolism and carbon metabolism were enriched, relative to the larval GNCs (Fig. S4C). Evaluation of the corresponding proteomic data largely confirmed the transcriptomic results. Again, oxidative phosphorylation and TCA cycle were significantly enriched in the 1-week-old GNCs, relative to the larval cells. In addition, KEGG pathways associated with carbohydrate, fatty acid, and amino acid metabolism were abound (Fig. S4C´). These data suggest that relative to larval GNCs, 1-week-old adult GNCs are characterised by a highly efficient energy production machinery, mainly driven by robust TCA cycle rates and extensive oxidative phosphorylation. A similar result was already obtained for the 1-week-old GNCs when compared with 1-week-old PNCs (Fig. 6).

To further assess age-dependent functional adaptations, we compared the adult PNC transcriptome and proteome at 3 and 5 weeks of age to those of 1-week-old cells. Principal component analysis (PCA) of the transcriptome data revealed a strong separation between the three age groups (Fig. 8A). However, at the level of individual transcripts we found more subtle differences between the 1- and 3-week-old cells (Fig. 8B), with merely 69 upregulated and 77 downregulated genes (FDR $p = <0.05$). Substantially more genes were differentially expressed after five weeks, relative to 1-week-old PNCs (197 upregulated, 278 downregulated, Fig. 8B). GO analyses of the genes upregulated after 3-weeks showed enrichment of the 'Metabolism of xenobiotics by cytochrome P450', 'Drug metabolism' and 'Glutathione metabolism' KEGG pathways, which involved three Glutathione S transferase genes (*GstD2*, *GstD3* and *GstT3*), the *Ribonucleoside diphosphate reductase large subunit* (*Rnrl*) and the *UDP-glycosyltransferase family 302 member K1* (*Ugt302K1*) (Fig. 8C). These genes remained upregulated after five weeks and thus appeared important for the PNCs during the ageing process. Although a GO analysis of the downregulated genes revealed no significantly affected KEGG pathways, downregulated biological processes

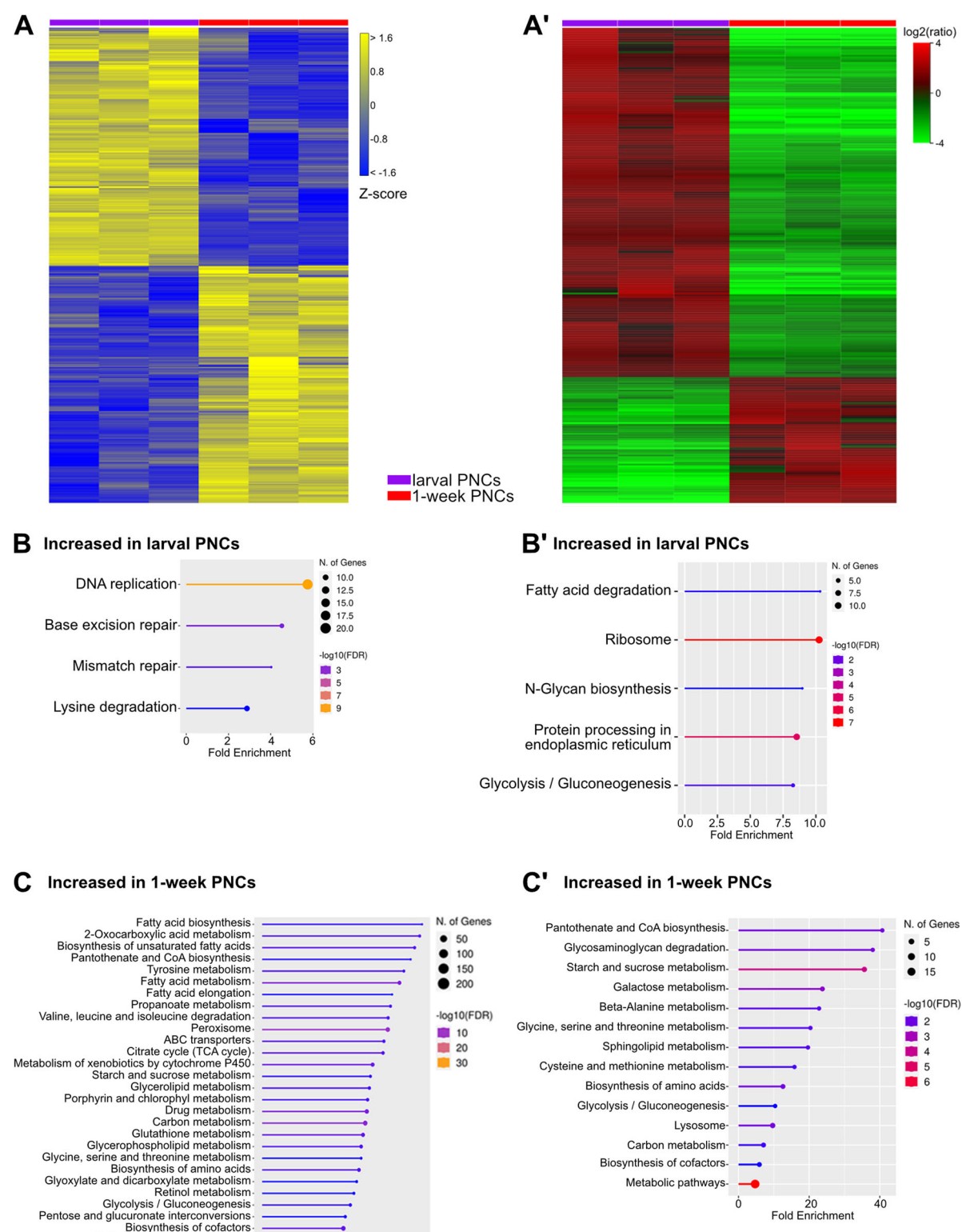

**Fig. 7 | Nephrocytes exhibit developmental stage-specific transcriptomic and proteomic signatures. A** Heatmap of RNAseq replicates (minimum fold change 1.5, FDR *p* value cutoff 0.05) from 3rd instar larval PNCs (purple bar) and 1-week-old adult PNCs (red bar). (A′) Heatmap of proteome replicates from 3rd instar larval PNCs (purple bar) and 1-week-old adult PNCs (red bar). **B** KEGG pathways (transcriptome-based) enriched in 3rd instar larval PNCs, relative to 1-week-old adult PNCs. (B′) KEGG pathways (proteome-based) enriched in 3rd instar larval PNCs, relative to 1-week-old adult PNCs. **C** KEGG pathways (transcriptome-based) enriched in 1-week-old adult PNCs, relative to 3rd instar larval PNCs. (C′) KEGG pathways (proteome-based) enriched in 1-week-old adult PNCs, relative to 3rd instar larval PNCs.

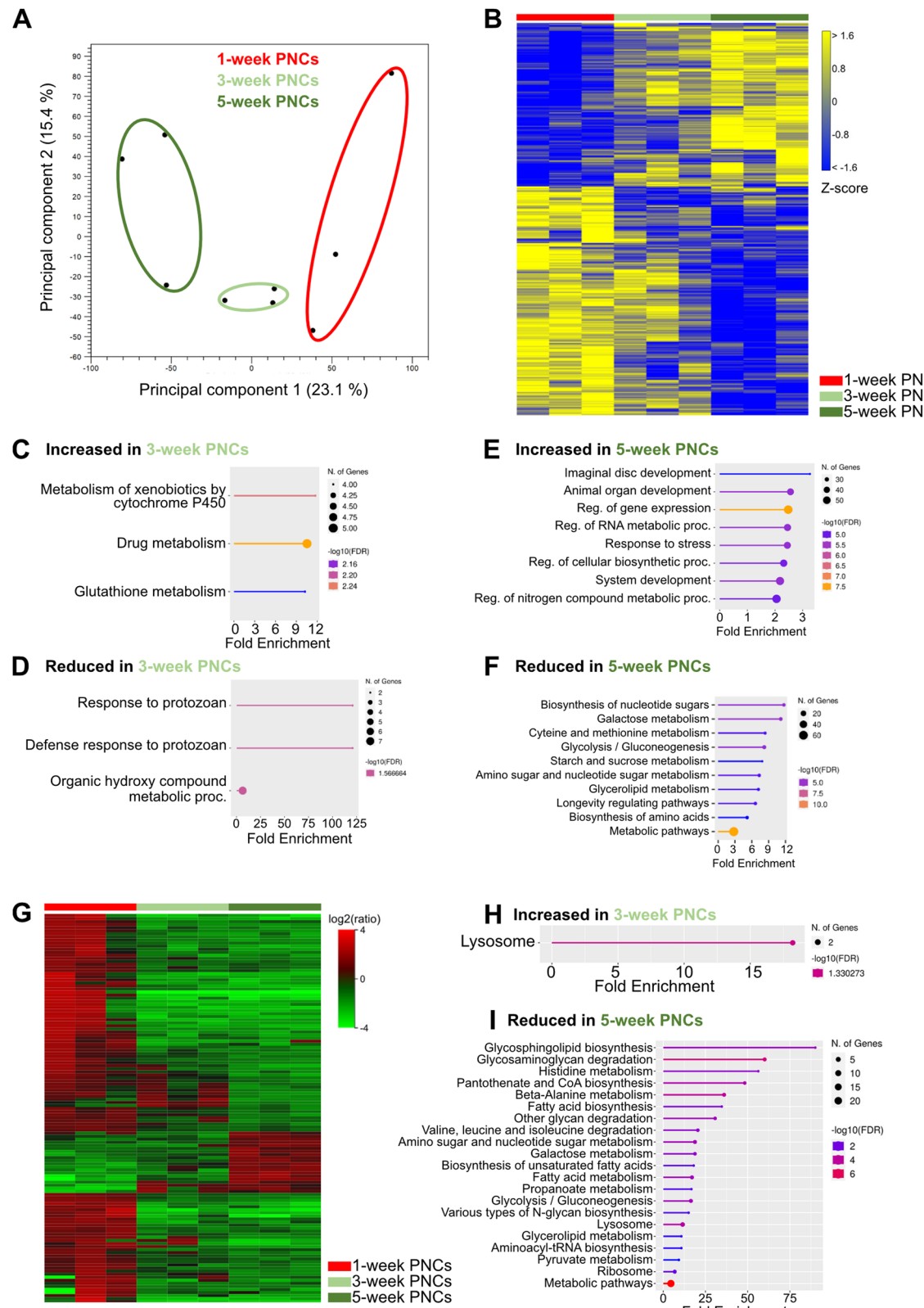

**Fig. 8 | Nephrocytes exhibit age-dependent changes in transcriptomic and proteomic signatures.** **A** Principal component analysis of transcriptome profile alterations among 1-week- (1w), 3-week- (3w) and 5-week- (5w) old PNCs. Circles were added manually. **B** Heatmap of RNAseq replicates (minimum fold change 1.5, FDR *p* value cutoff 0.05) from 1-week-old PNCs (red bar), 3-week-old PNCs (light green bar) and 5-week-old PNCs (dark green bar). **C** KEGG pathways enriched in 3-week-old PNCs, relative to 1-week-old PNCs (transcriptome-based). **D** KEGG pathways (transcriptome-based) reduced in 3-week-old PNCs, relative to 1-week-old PNCs. **E** Biological processes increased in 5-week-old PNCs, relative to 1-week-old PNCs (transcriptome-based). **F** Biological processes reduced in 5-week-old PNCs, relative to 1-week-old PNCs (transcriptome-based). **G** Heatmap of proteome replicates isolated from 1-week-old PNCs (red bar), 3-week-old PNCs (light green bar) and 5-week-old PNCs (dark green bar). **H** KEGG pathways enriched in 3-week-old PNCs, relative to 1-week-old PNCs (proteome-based). **I** KEGG pathways reduced in 5-week-old PNCs, relative to 1-week-old PNCs (proteome-based).

relating to 'Defense response to protozoan' (relevant genes: *l(2)34Fc* and *Drs*) and 'Organic hydroxy compound metabolic process' (relevant genes: *Inos*, *Drat*, *CG2070*, *Cyp4aa1*, *Ip6k*, *PPO1* and *PPO2*) were identified (Fig. 8D). The enriched biological processes at five weeks in the upregulated group were related to development and regulation of gene expression but also to stress responses (Fig. 8E), whereas the KEGG pathways identified from the downregulated genes were associated with carbohydrate, amino acid, and lipid metabolism (Fig. 8F). In addition, genes related to the 'Longevity regulating pathway' were enriched in this group, including several heat shock protein (Hsp) genes (*Hsp68*, *Hsp70Ab*, *Hsp70Bb*, *Sod3*, *Hsp70Bbb*, *alc*, and *Hsc70-4*).

The changes in the proteome after three weeks, compared to 1-week-old PNCs, were minimal (Fig. 8G and Supplementary Data 5A, B), and the corresponding KEGG pathway analysis revealed that only the 'Lysosome' pathway was significantly enriched in 3-week-old PNCs. This result was mainly based on the elevated levels of two hydrolases (CathD, CG7997) in the respective cells (Fig. 8H and Supplementary Data 5C). Somewhat divergent from the transcriptome results, the proteomic dataset indicated a reduced ability of older PNCs (5 weeks) to degrade glycans and proteins when compared to 1-week-old PNCs (Fig. 8I and Supplementary Data 5D). We identified that the main factor responsible for this age-dependent decline in glycan and protein degradation capacity was the reduced abundance of acid hydrolases (e.g., Hexo1, Hexo2, and Ect3; Supplementary Data 5D). Moreover, 1-week-old PNCs exhibited proteomic signatures indicative of increased pantothenate and coenzyme A (CoA) biosynthesis (increased levels of Bcat, CG5618, and CG31075, Fig. 8I and Supplementary Data 5D). CoA is an essential cofactor for numerous cellular oxidative pathways, including amino acid oxidation; therefore, its increased levels in 1-week-old PNCs and the expected abundance of amino acids derived from protein degradation could indicate that these cells rely on amino acid oxidation as an energy source. The enrichment of the amino acid degradative pathways in the proteome of 1-week-old PNCs (increased levels of relevant enzymes, e.g., Bcat and Hibch; Supplementary Data 5D), relative to 5-week-old cells (Fig. 8I), supports this possibility. Pathways with putatively increased activities in 5-week-old PNCs, compared to their 1- and 3-week-old counterparts, were not detected using the applied significance criteria.

In contrast to the PNC profiles, the GNC proteomes did not indicate a reduced ability of 5-week-old cells to degrade glycans or proteins (relative to 1-week-old GNCs). The older cells were rather characterised by a significant increase in the 'Lysosome' KEGG pathway, which was based on elevated levels of the protease Cathepsin B1 and V-ATPase subunit 'a' (Vha100-2, Fig. S2 and Supplementary Data 6A–C). Thus, in terms of ageing, an inverse trend may exist for the two types of *Drosophila* nephrocytes, whereby the PNCs progressively lose their ability to degrade sequestered material from the haemolymph, while the GNCs gradually increase this capacity.

During these experiments, we identified substantial differences in the transcriptomes and proteomes of 5-week-old PNCs, relative to 1-week-old PNCs. On the other hand, the differences between 3-week-old PNCs and 1-week-old PNCs were rather subtle (Fig. 8C, D, H). These results indicate that most of the functionally relevant effects of ageing are restricted to cells older than three weeks, while younger cells maintain a largely constant functionality.

## Discussion

*Drosophila* nephrocytes are highly adapted cells that are essential for sequestering and storing toxic or non-metabolisable materials from the haemolymph. Based on their considerable structural and functional similarities to vertebrate podocytes, these cells are preferred models for assessing the molecular bases of distinct types of human nephropathies[22,28,47]. In this study, we applied a combination of transcriptomic and proteomic analyses to determine the predominant molecular functions but also identify the functional differences between the two types of *Drosophila* nephrocytes (GNCs and PNCs). In addition, the influence of developmental stage and

ageing on the transcriptomic and proteomic signatures of the cells was assessed. Our studies proved that the *Drosophila* nephrocytes fulfil additional functions beyond their activity as scavenger cells.

We established an experimental setup that allowed us to circumvent challenges in previous *Drosophila* nephrocyte transcriptome and proteome analyses. In a recent study, GFP-labelled Malpighian tubules and nephrocytes were dissected and processed for single-nucleus RNA sequencing to decipher the transcriptome of 'fly kidney' cells[30]. In line with our data, the results of that study indicated that garland and pericardial nephrocytes are not as interchangeable as previously considered. Interestingly, a cross-species analysis revealed that the transcriptome of *Drosophila* adult PNCs exhibits considerable similarities with the transcriptome of mouse podocytes. In contrast, the *Drosophila* adult GNC transcriptome is more similar to the mouse parietal epithelium transcriptome[30]. The functional difference suggested by these data was corroborated by our results, which indicate, among other things, a higher endocytic and degradative capability of PNCs compared to GNCs (Fig. 5) as well as a significantly different cellular energy metabolism (Fig. 6). Of note, the authors of the former study found that the garland and pericardial nephrocytes were underrepresented in their dataset[30], likely because GNCs are binucleated and because PNCs harbour large nuclei (polyploidisation). A similar phenomenon was reported for vertebrate kidney cells[48–50]. Thus, garland and pericardial nephrocytes appear somewhat ill-suited for automated cell sorting and single-nucleus sequencing. Therefore, manual cell sorting was used in this study to ensure homogeneity of the individual cell populations. Of note, the protocol used herein required collagenase treatment, which must be considered when evaluating the results obtained. Collagenase-based degradation of the basement membrane not only facilitates cell isolation, but may also release growth factors and chemokines trapped within the ECM mesh. In addition, structural ECM components are probably also released from the cell surface, making them difficult to detect after cell isolation. Nevertheless, given the robust, cell-type-specific results we have obtained, our method qualifies as an efficient complement to current high-throughput approaches.

### Garland and pericardial nephrocytes exhibit distinct transcriptomic and proteomic signatures

When comparing the two populations of *Drosophila* nephrocytes, it is evident that both cell types share a set of critical features. These include the expression of characteristic genes such as *sns* and *duf*, encoding constituents of the slit diaphragm[28], *hand* and *Klf15*, encoding cell fate determinants[20,31], and *Amnionless* and *Cubilin*, coding for proteins which facilitate receptor-mediated endocytosis[36]. However, whether these two populations of nephrocytes are indeed functionally equivalent or rather fulfil unique physiological functions in the animal has not yet been explored in detail. By evaluating transcriptomic and proteomic signatures of 1-week-old nephrocytes and comparing them to complete animals, we found evidence that both cell types were characterised by elevated vesicle-mediated transport and proteoglycan degradative capacities, supporting the well-described scavenger function of both cell populations. However, when comparing the profiles of GNCs to those of PNCs, the latter indicated a higher glycan and protein degradative capability and an elevated endocytosis and phagolysosome assembly efficiency (Fig. 5B, B′). These results imply that PNCs are more efficient in degrading sequestered materials than GNCs. This difference was most pronounced in young cells (1 week of age, Fig. 5 and Supplementary Data 2B) and appeared to decrease with age because of the progressive loss of factors required to degrade sequestered material from the haemolymph in PNCs, while the GNCs increased this capability over time (Fig. S2 and Supplementary Data 6C). Thus, regarding degradative capacity, the two types of *Drosophila* nephrocytes appeared to show a somewhat inverse trend during ageing, with greater PNC activity in younger cells, while GNCs increased their activity in older animals.

Moreover, PNCs exhibited transcriptomic and proteomic signatures, which suggested a substantially decreased oxidative phosphorylation (OXPHOS) rate, while GNCs were characterised by elevated levels of transcripts and proteins that ensure efficient OXPHOS and TCA-cycle

| Process | 3rd larva PNCs | 3rd larva GNCs | 1w PNCs | 1w GNCs | 5w PNCs | 5w GNCs |
|---|---|---|---|---|---|---|
| Endocytosis | ++ | + | ++ | + | + | + |
| Protein / glycan degradation | o | + | ++ | + | + | ++ |
| Protein / glycan biosynthesis | ++ | + | - | - | o | o |
| ECM | o | + | ++ | ++ | ++ | ++ |
| OXPHOS | o | o | - | ++ | o | ++ |
| TCA cycle | o | o | - | ++ | o | ++ |
| Immune signalling | o | o | + | + | + | + |

**Fig. 9 | Summary of garland and pericardial nephrocyte characteristics.** Depicted are processes that were indicated as significantly upregulated (+, ++) or downregulated (−) in GNCs or PNCs, relative to complete animals (o = not significantly altered). '1w' and '5w' indicate animals of 1 week and 5 weeks of age, respectively.

progression (Figs. 6 and S1 and Supplementary Data 2C, 3B). Utilising a DCFDA-based reactive oxygen species (ROS) detection assay, we found substantially increased amounts of ROS in GNCs, relative to PNCs. This result likely reflected the elevated OXPHOS rate in the GNCs, and thus supported the high physiological relevance of our transcriptomic/proteomic datasets (Fig. 6C, C'). Hence, the two types of nephrocytes appeared to rely on basically different means of energy production.

Notably, the reduction in factors required for oxidative phosphorylation in PNCs was significant compared to both GNCs and complete animals (Fig. 2H and Supplementary Data 1C, 2C), implying that PNCs rely on a rather atypical form of ATP production. In this regard, the fact that during the process of oxidative phosphorylation, electrons leak and subsequently interact with molecular oxygen to form superoxide or other ROS may be relevant[44–46]. It has been shown that the amount of ROS produced in PNCs significantly influences cardiac function, which is based on ROS-induced activation of downstream MKK3-p38 signalling in PNCs as opposed to the extracellular distribution of ROS from the PNCs into cardiomyocytes. This signalling has been suggested to modulate the expression of pericardial adhesion molecules in contact with adjacent cardiomyocytes, which consequently affects cell-cell adhesion between PNCs and cardiomyocytes and impacts heart function[51]. In addition, PNC intrinsic ROS production appears to activate a signalling cascade transduced by Ask1, c-Jun N-terminal kinase, and p38, which induces expression of the cytokine Unpaired 3 (Upd3). In turn, the Upd3 released by the PNCs controls the expression of the cardiac extracellular matrix (cECM) protein Pericardin, which affects cECM composition, and consequently, cardiac function and lifespan[52]. The functional relevance of Pericardin to proper heart assembly and function has been shown previously[53]. These results introduce PNC-derived ROS as metabolic signals affecting cECM composition and heart function and thus emphasise the physiological need for PNCs to tightly control ROS production.

In addition to their relevance in paracrine signalling, high levels of ROS are harmful to cells because they cause the oxidation of numerous biological molecules, including lipids, proteins, and DNA[54–56]. To circumvent this cellular damage and to ensure long-term functionality without cell renewal, PNCs may rely on means of ATP production that include only low rates of oxidative phosphorylation, which correspond to low ROS production rates. As opposed to oxidative phosphorylation, 1-week-old PNCs may utilise extensive glycolysis as a means of energy production, with the necessary glucose levels provided through elevated gluconeogenesis. The required glucogenic amino acids could be derived from degraded proteins, which are generally abundant in PNCs. The physiological relevance of this energy production option is supported by transcriptomic and proteomic signatures that indicate both increased gluconeogenesis as well as increased amino acid degradation rates in 1-week-old PNCs (Figs. 2G, 3B, 5A' and 7C, C'). The reduced but still existent ROS formation may become increasingly

detrimental with age, eventually resulting in the structural and functional demise of PNCs, which occurs in older animals[16].

In addition to adult PNC analyses, we assessed corresponding cells from 3rd instar larval tissue. The transcriptome and proteome profiles of the larval PNCs indicated a significantly reduced ability to degrade amino acids and glycans and a reduced abundance of lysosomal acid hydrolases relative to 1-week-old adult PNCs (Fig. 7C, C' and Supplementary Data 4C). This result suggests that the pericardial nephrocytes attain their full ability to degrade compounds scavenged from the haemolymph only in the adult stage. However, in contrast to adult PNCs, the larval cells revealed proteomic signatures that indicated effective protein biosynthesis mechanisms, particularly regarding proteins that require N-glycosylation and additional processing within the ER (Fig. 7B' and Supplementary Data 4B). Given that N-glycosylation is a critical maturation step for most secreted proteins[57,58], a primary function of larval PNCs appears to include the production and secretion of distinct proteins, many of which likely represent haemolymph circulating factors. The significant differences between garland and pericardial nephrocytes and their stage and age-dependent characteristics are displayed in Fig. 9. Volcano plots depicting all comparisons performed in this study are shown in Fig. S6.

## The transcriptomic signatures of garland and pericardial nephrocytes indicate elevated immune signalling

While altered immune activity was not significantly detected at the proteome level, it was considerably enriched in PNCs and GNCs at the transcriptome level relative to complete animals. In addition, a corresponding increase was also evident in adult PNCs compared to the larval cells. Enrichment of the Toll and Imd pathways was found to be one of the key characteristics of both types of nephrocytes (Fig. 2D). Further analysis revealed the increased expression of several components of both pathways (Fig. 4). It has been reported that nephrocytes remove Lys-type peptidoglycan (PGN) from the haemolymph by filtration. This scavenging activity limits Toll-dependent systemic immune activation. Animals without nephrocytes or with reduced nephrocyte function show increased survival to infection, likely due to an excess of PGN in the haemolymph and resulting constitutive activation of the Toll pathway[43]. Our data revealed moderately enriched expression of several Toll pathway components (extra- and intracellular) in both PNCs and GNCs. Besides the important function of PGN filtration from the haemolymph, nephrocytes are exposed to pathogens during systemic infection. To avoid being damaged by invading pathogens, they need a functional immune recognition and defense system. Peptidoglycan recognition receptors (PGRPs) with amidase activity are not relevantly expressed in either type of nephrocyte, compared to the whole animal. However, the PGRPs of the Imd pathway (*PGRP-LC, -LE* and *-LA*) are highly enriched,

as is the downstream transcription factor *Relish*. The Imd pathway is strongly regulated by factors such as PGRP-LF, pirk, and caudal (Fig. 4). While *caudal* showed no enrichment in PNCs and GNCs, *PGRP-LF* and *pirk* expression was highly enriched. Based on the assumption that nephrocytes only remove Lys-type PGN from the haemolymph[43], and not DAP-type PGN, which is normally recognised by PGRP receptors, it is likely that the remaining PGN in the haemolymph is required to avoid overactivation of the pathway. Despite this tight regulation that inhibits unwanted activity of the Imd-pathway in nephrocytes, this pathway likely ensures their survival during systemic infection. Thus, our data support the important role of nephrocytes in immune homoeostasis and suggest a strong capacity of these cells to mount their immune response against gram-positive and gram-negative bacteria. To demonstrate that the nephrocytes are immune-competent, we performed infection experiments by injecting bacteria. Here, the bacteria-specific induction of *Drs-GFP* expression, which is the most important antimicrobial peptide gene in a systemic infection, confirmed our assumptions. Therefore, we propose to include nephrocytes in the list of immune-competent organs/cells in *Drosophila*.

## An extended perspective on garland and pericardial nephrocytes
Our results show that the previous view of nephrocytes was rather incomplete and their physiological functions were only partially known. Existing literature suggests that garland and pericardial nephrocytes are two populations of the same cell type, with basically identical functionality—to act as scavenger cells. However, our data indicate that this is only partially true. Although both cell types share overlapping activities in the endocytosis and degradation of metabolites and other substances that need to be removed from the haemolymph, they exhibit striking differences in their energy metabolism and probably secrete a different spectrum of proteins. The atypical form of energy production in PNCs is likely connected to the close association of the cells to cardiomyocytes and the high sensitivity of the contractile heart tissue to ROS signals.

Regarding the scavenger function of the cells, except for one candidate protein, we did not identify any haemolymph circulating factors in the nephrocyte proteome that were not simultaneously present in the corresponding transcriptome. This is an interesting observation since respective factors can be considered as being sequestered from the haemolymph, rather than being expressed by the nephrocytes. The almost complete absence of such factors suggests that nephrocytes rapidly degrade the endocytosed proteins and do not store them. The only exception we have found is the yet uncharacterised protein CG6409 (Fig. 3). Given the lack of experimental data on the molecular function of this protein, it remains unclear whether it is stored by the nephrocytes or just degraded rather slowly.

These data along with the apparent relevance of PNCs and GNCs in immune signalling extend the current knowledge on nephrocyte function in *Drosophila* and can represent a valuable basis for a more specific application of this model in the analysis of renal cell function in health and disease.

## Methods
### Fly strains
*hand*C-GFP, and *hand*C-mCherry have been described previously[31,59]. *Klf15*[NN] flies were kindly provided by Paul Hartley[20]. The antimicrobial peptide reporter lines were constructed by Tzou et al.[40] *w*[1118] wildtype flies were purchased from Bloomington *Drosophila* Stock Center (BDSC_5905).

### Isolation of garland and pericardial nephrocytes
After anaesthetisation, female larvae or adult flies (*hand*C-GFP) were transferred to a small Sylgard-coated petri dish and dissected in PBS buffer. For larvae, insect pins were used to fix the specimens ventral side up at the anterior and posterior ends as they were completely covered with PBS buffer. The cuticle was cut longitudinally with spring scissors and fixed with insect pins to facilitate the removal of the viscera, which was performed with

forceps until the heart tube and proventriculus were visible. Adult flies were placed ventral side up and fixed by an insect needle through the thorax for pericardial nephrocyte (PNC) isolation or through the abdomen for garland nephrocyte (GNC) isolation. After covering the specimens with PBS, the legs were detached and a single cut with the spring scissors removed the ventral side of the abdomen (for PNC isolation) or the thorax, including the head (for GNC isolation). To expose the heart, the viscera and excess fat bodies were removed with forceps. Both sides of the abdomen were then attached to the Sylgard with insect pins.

The PBS covering the dissected specimens was replaced by PBS containing 1 mg/ml collagenase (#C0130, Sigma-Aldrich, Darmstadt, Germany). Samples were incubated for 15 min at 37 °C. Subsequently, the collagenase was diluted by carefully adding PBS buffer. The petri dish was then placed under a fluorescence stereo microscope to identify the GFP-labelled nephrocytes (*hand*C-GFP flies for transcriptomic and proteomic analysis; *hand*C-mCherry for ROS measurements (see below)).

A microinjector (CellTram®4rOil, Eppendorf, Hamburg, Germany) equipped with a custom-made glass capillary was used to extract individual nephrocytes (for numbers of isolated cells, see Table 1) from the collagenase-treated ECM meshwork and directly transfer them into Roti®Zol (Roth, Karlsruhe, Germany) for transcriptome analysis or lysis buffer for proteome analysis (iST sample preparation kit, Preomics, Planegg, Martinsried, Germany).

### Whole animal protein extracts
For the extraction of proteins from whole animals, five female *hand*C-GFP adult flies (aged 1 week) were collected and placed in a reaction cup. Subsequent to the addition of 120 µl of lysis buffer (iST sample preparation kit), the flies were homogenised with a pestle, centrifuged at $10,000 \times g$, 4 °C, for 10 min, and the resulting supernatant was placed into a thermomixer operating at 1000 rpm, 95 °C, for 10 min (according to the manufacturer's protocol). The subsequent processing of the samples was performed according to the procedures specified under 'Sample preparation for proteomic analyses'.

### ROS measurements in nephrocytes
ROS measurements were done with a reactive oxygen species (ROS) detection assay kit (ab287839, Abcam, Oxford, UK). Briefly, 1-week-old female *hand*C-mCherry flies were dissected in artificial haemolymph[60], and nephrocytes were isolated as described (see 'Isolation of garland and pericardial nephrocytes'). Isolated cells were transferred onto a poly-l-lysin (#P4707, Sigma-Aldrich, Darmstadt, Germany) coated coverslip and moistened with artificial haemolymph. Subsequently, the haemolymph was replaced by 2× ROS label diluted in artificial haemolymph. Cells were incubated for 30 min at 28 °C in the dark. The buffer was removed and nephrocytes were fixed for 5 min with 8% formaldehyde in artificial haemolymph, followed by a short washing step with artificial haemolymph. Nephrocytes were then embedded in Fluoromount-G mounting medium containing DAPI (#00-4959-52, Thermo Scientific™, Waltham, MA, USA) and single slice images were captured immediately using an LSM800 (Zeiss, Jena, Germany) equipped with fluorescein and DAPI emission filters. As a negative control, cells were treated similarly but without ROS label reagent. Images were analysed for ROS signal intensity and the number of ROS-positive punctae.

The 'mean pixel intensity' measurement function provided by the Fiji software package was used to quantify the ROS signal intensity for each cell. To subtract the background signal, the mean of the 'mean pixel intensity' of the negative control for PNCs and GNCs, respectively, was calculated. The corresponding value was subtracted from the mean pixel intensity value of each measured PNC or GNC.

For the analysis of ROS-positive punctae, the 'MaxEntropy' auto threshold, implemented in Fiji, was used to highlight ROS-positive punctae above the given set threshold. Size and number were analysed using the 'analyze particle' function. The number of ROS-positive punctae within a certain cell area was calculated. A Mann–Whitney *U*

test was used for statistical analysis using GraphPad Prism 9 (Boston, MA, USA).

## MitoTracker staining in nephrocytes

1-week-old female *hand*C-mCherry flies were dissected in Shields and Sang M3 Insect Medium (#S8398, Sigma-Aldrich, Darmstadt, Germany) supplemented with 20% FBS (#P30-1402, PAN-Biotech, Aidenbach, Germany) (hereinafter abbreviated as M3 Media) and 1 mg/ml Collagenase followed by an incubation step for 15 min at 30 °C. Afterwards, garland and pericardial nephrocytes were isolated and transferred onto poly-l-lysin coated coverslips moistened with M3 Media containing 500 nM MitoTracker™ Red CMXRos (#M7512, Thermo Scientific™, Waltham, MA, USA). Nephrocytes were incubated for 45 min at 30 °C in the dark. The staining solution was removed and nephrocytes were embedded in Fluoromount-G mounting medium (#00-4958-02, Thermo Scientific™, Waltham, MA, USA), and single slice images were captured using an LSM800 (Zeiss, Jena, Germany) equipped with MitoTracker Red and GFP emission filters. For the analysis of the MitoTracker Red positive area, the 'substract background' function provided by the Fiji software package was used (Rolling ball radius: 50 pixel and 'disable smoothing'). A threshold was set to highlight 15% of the brightest pixels for each cell. The MitoTracker Red positive area was then determined using the 'analyze particle' function and set in relation to the cell area (without the nuclei area). A two-tailed, unpaired t-test was performed for statistical analysis using GraphPad Prism 9 (Boston, MA, USA).

## Avidin-Cy3 Uptake assay

Larvae, 1-week, and 5-week old female *hand*C-GFP flies were dissected in artificial haemolymph as described in 'Isolation of garland and pericardial nephrocytes'. The preparation buffer was replaced by artificial haemolymph containing 0.02 mg/ml Avidin-Cy3 (#E4142, Sigma-Aldrich, St. Louis, Missouri, USA). Specimens were incubated for 5 min at RT in the dark in the staining solution followed by a short washing step with artificial haemolymph. Uptake was stopped by fixation with 4% methanol-free formaldehyde in artificial haemolymph. After two short washing steps with artificial haemolymph, the specimens were embedded in Fluoromount-G mounting medium and images of PNCs were captured using identical settings. The mean pixel-intensity measurement function provided by the Fiji ImageJ software package was used to quantify uptake efficiency. Pixel intensity was measured in relation to the perimeter of the cell. For statistical analysis, a one-way ANOVA was performed (GraphPad Prism 9).

## Systemic infection with *Pectobacterium carotovorum*

Bacteria (*Pectobacterium carotovorum*) were cultured in LB broth at 30 °C overnight. Bacteria were concentrated to an $OD_{600}$ of 500 by centrifugation and removal of excess LB broth. Five-day-old animals of the *Drs*-GFP and *Def*-GFP reporter lines were either not pricked in the thorax, pricked with a needle dipped in LB broth (sterile), or pricked with a needle dipped into the concentrated bacteria. PNCs were monitored 24 h after infection. The exposure time was kept constant through the groups compared (*Drs*-GFP = 350 ms; *Def*-GFP = 800 ms).

## RNA extraction, cDNA synthesis and amplification

RNA extraction with Trizol was performed as described previously[61]. cDNA synthesis and amplification of the whole-length cDNA was performed as described previously, with slight modifications, and adjusted to a 10 μl reaction volume[62]. An Oligo dT primer and cap finder primer that bound the 5′-cap of mRNA were used along with LA Taq polymerase (Takara Bio Inc., Kusatsu, Shiga, Japan) for amplification. After optimising the cycling protocol and determining a cycling number in the exponential phase, the corresponding parameters were used. The amplified full-length cDNA was cleaned using the Monarch PCR & DNA clean-up Kit (New England Biolabs, Ipswich, MA, USA) following the manufacturer's protocol. DNA concentrations were determined using a Qubit fluorometer with a high-sensitivity dsDNA kit (Thermo Fisher Scientific, Waltham, MA, USA).

## Library preparation and sequencing

Library preparation was conducted using a protocol adapted from[63]. The samples were equimolar pooled and quantified (Agilent Bioanalyzer, DNA 7500). Sequencing was performed using a NextSeq 500 (Illumina) with the High Output Kit v2.5 (75 cycles).

## RNA-Seq data processing

Differential expression analysis of the RNA sequencing data was performed using the CLC Genomics Workbench software (Qiagen, Hilden, Germany). Reads were mapped to the BDGP6 reference with the RNA-Seq Analysis tool, and differential expression was calculated with the Differential Expression for RNA-Seq tool. Differentially expressed genes (DEGs) were filtered by false discovery rate (FDR) ($p$ = <0.05) and a fold change cut-off of 1.5.

## Sample preparation for proteomic analyses

The manually isolated cells were collected in lysis buffer (iST sample preparation kit, PreOmics, Martinsried, Germany) and further processed for mass spectrometry analyses using the 'iST Kit' (PreOmics) according to the manufacturer's instructions. Briefly, solubilisation, reduction, and alkylation were performed in sodium deoxycholate (SDC) buffer containing TCEP and 2-chloroacetamide. Proteins were enzymatically hydrolysed overnight at 37 °C by adding LysC and trypsin. Peptides were de-salted, dried by vacuum centrifugation and reconstituted in 20 μl 0.1% trifluoroacetic acid.

## Mass spectrometry and data analysis

Reversed-phase chromatography was performed using an UltiMate 3000 RSLCnano System (Thermo Fisher Scientific, Waltham, MA, USA). Samples were loaded onto a trap column (Acclaim PepMap 100 C18, 5 μm, 0.1 mm × 20 mm, Thermo Fisher Scientific) and washed with a loading buffer (0.1% TFA in H2O) at a flow rate of 25 μl/min. The trap column was switched in line with a separation column (Acclaim PepMap 100 C18 2 μm, 0.075 mm × 150 mm, Thermo Fisher Scientific). Bound peptides were eluted by adjusting the mixture of buffer A (99% water, 1% acetonitrile and 0.1% formic acid) and buffer B (80% acetonitrile, 20% water and 0.1% formic acid) from 100:0 to 20:80 over 60 min. The flow rate was maintained at 0.3 μl/min. The eluted compounds were directly electrosprayed through an EASY-Spray ion source (Thermo Fisher Scientific) into a Q Exactive Plus Orbitrap mass spectrometer (Thermo Fisher Scientific). The eluates were then analysed by measuring the intact molecule and fragment masses, generated by higher-energy collisional dissociation (HCD) of the corresponding parent ion. To determine peptide-specific amino acid sequences (parent mass error tolerance: 10 ppm; fragment mass error tolerance: 0.2 Da; enzyme: trypsin; max missed cleavages: 2; selected PTMs: carbamidomethylation, oxidation, and phosphorylation), PEAKS Studio software (Version 10.6, Bioinformatics Solutions Inc., Waterloo, Canada), in combination with a *Drosophila*-specific database (UP000000803, www.uniprot.org/proteomes/UP000000803), was used. Label-free quantification was conducted by comparing peptide and protein amounts of different groups according to established protocols[64], with each group consisting of a minimum of three independent biological replicates. The protein list was controlled by FDR (threshold: 1%) and significance was calculated using PEAKS software (one-way ANOVA). Classification as a relevant factor required a $p$ value of <0.01, with quantification based on a minimum of two individual protein-specific peptides. The mass spectrometry proteomics data have been deposited to the ProteomeXchange Consortium via the PRIDE partner repository (http://www.ebi.ac.uk/pride [65]) with the dataset identifier PXD047262.

## Gene Ontology (GO) analysis

A GO enrichment analysis was performed using ShinyGO, v0.77[66]. The following parameters were applied: FDR cut-off: 0.05; # of pathways to

show: 30; pathway size (min): 2; pathway size (max): 2000. The Kyoto Encyclopedia of Genes and Genomes (KEGG)-pathway analysis was based on the KEGG pathway database[67].

## Statistics and reproducibility

For all experiments, statistical analysis was performed with GraphPad Prism 9 (GraphPad Software Inc., SanDiego, CA, USA). The unpaired two-tailed Student's $t$ test was used for the analysis of differences between the two groups. One-way ANOVA followed by a Turkey posthoc test was used for the multiple comparisons. Two-way ANOVA was performed to analyse two different categorical independent variables on one continuous dependent variable. All values of histograms were means ± SD. For all boxplots, the centre line of a plot indicates the median; the upper and lower bounds indicate the 75th and 25th percentiles, respectively; and the whiskers indicate the minimum and maximum. The proteomics data were statistically analysed and visualised with PEAKS Studio software. All experiments were performed at least in triplicates.

## Reporting summary

Further information on research design is available in the Nature Portfolio Reporting Summary linked to this article.

## Data availability

The data supporting the findings from this study are available within the manuscript and its Supplementary Information. The mass spectrometry proteomics data have been deposited to the ProteomeXchange Consortium via the PRIDE partner repository (http://www.ebi.ac.uk/pride) with the dataset identifier PXD047262. SwissProt database (UP000000803, www.uniprot.org/proteomes/ UP000000803) was used to determine peptide-specific amino acid sequences. The transcriptomic data have been deposited to the Gene Expression Omnibus database (https://www.ncbi.nlm.nih.gov/geo) with the dataset identifier GSE250029.

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

## Acknowledgements

We thank Martina Biedermann, Eva Cordes, Kerstin Etzold, Mechthild Krabusch, Britta Laubenstein and Christiane Sandberg for excellent technical assistance, and Stefan Walter for supporting mass spectrometry analyses. This work was supported by grants from the Deutsche Forschungsgemeinschaft to A.P. (PA 517/12-1, 12-2; SFB 944, TP7 and SFB 1557, TP12), T.R. (SFB 1182, TP2), H.M. (HA 6421/4-1; SFB 944, TP21), and by the Open Access Publishing Fund of Osnabrück University.

## Author contributions

Conceptualisation: H.M., J.B., T.R., and A.P.; Investigation: H.M., J.B., M.J., and SK;. Resources: X.M.; Data curation: J.B., H.M., M.J., and S.K.; Writing—original draft: H.M., J.B., T.R., and A.P.; Writing—review & editing: H.M., J.B., M.J., X.M., S.K., T.R., and A.P.; Visualisation: H.M., J.B., and M.J.; Supervision: T.R. and A.P.; Project administration: H.M., J.B., T.R., and A.P.; Funding acquisition: H.M., T.R., and A.P.

## Funding

## Competing interests

The authors declare no competing interests.
