## [Peer Review file · Communications Biology]

Combined transcriptome and proteome profiling reveal cell-type-specific functions of *Drosophila* garland and pericardial nephrocytes

Corresponding Author: Professor Achim Paululat

Version 0:

Reviewer comments:

Reviewer #1

(Remarks to the Author)

Meyer et al. characterize both types of *Drosophila* nephrocytes employing mass spectrometry based proteomics and RNA Seq based transcriptomics. While both types of nephrocytes exhibit structures similar to mammalian slit-diaphragms little is known whether garland and pericardial nephrocytes fulfil distinct functions. The authors now characterize both cell types at several time points at one, three and five weeks as adults and pericardial nephrocytes also at the 3rd instar larval stage. Functional enrichment analysis suggested that larval and adult nephrocytes, as well as garland and pericardial nephrocytes, fulfil distinct physiological functions. Pericardial nephrocytes were characterised by transcriptomic and proteomic profiles suggesting an atypical energy metabolism with very low oxidative phosphorylation rates. Moreover, the nephrocytes displayed signatures indicative of an extensive immune signalling.

Major points:

While this study characterizes differential gene and protein expression of the two different nephrocytes populations and makes it possible to formulate hypotheses about their distinct functions during the life of a fly, the data is descriptive, and changes found in gene expression were mostly not further characterized and confirmed with other methods.

In order to show the relevance of the genomics and proteomics data, it is important to focus on one pathway and delve in deeper by performing knockdown experiments analysing a potential phenotype (e.g. slit-diaphragm integrity). It would be highly interesting to dissect the functional implications of a subset of differentially expressed genes, e.g. the regulation of immune pathways in the respective nephrocytes. In my opinion showing functional implications of one of the identified targets is a prerequisite.

Altogether, the study is interesting and novel to the field, but needs further investigation.

Minor points:

Table 1: Why were transcriptomics and proteomics performed with 3rd instar larvae PNCs but not GNCs, for a full characterization and understanding of the aging process of both cell types the analysis of 3rd instar larvae GNCs might be interesting.

Fig.2 F: Are there also differences in the proteome analysis of 1 week old GNCs compared to total fly?

Fig.4 B: As Mtk is the only gene that is highly expressed in GNCs an analysis of Mtk GFP-reporter in GNCs might be interesting.

Fig.6B: Please name the respiratory chain complexes in the figure for better understanding, e.g. NADH dehydrogenase, etc.

Fig. 7B, C, B', C': Please increase the size of the labeling.

Fig. 8: Please increase the size of the labeling for C,D,E,F,H, I

Fig.9: Please sort the columns with increasing age, e.g. starting with the larval stage, for easier perusal.

Discussion: Please discuss the impact of the collagenase treatment on nephrocytes and the basement membrane and potential impact this could have had on the results.

Reviewer #2

(Remarks to the Author)

The manuscript by Meyer et al. delves into the distinct subtypes of *Drosophila* nephrocytes across various developmental stages through transcriptome and proteome analyses. They adopt a meticulous approach of manually sorting these cells to ensure proper sample homogeneity. In that manner, the authors lay important groundwork for this podocyte model, nicely confirming that pericardial and garland nephrocytes are clearly distinct cell types with divergent metabolism. The authors identify two candidates for the poorly understood secretory function of nephrocytes and propose an antimicrobial role for nephrocytes. However, the data presentation should be improved. Trying to tackle many aspects, additional experimental data to support the conclusions drawn from the high throughput data remains fairly limited.

- The last section of the introduction on page 5 reads more like the first part of a results section.
- Fig. 2A: 30 cells per measurement in GCN seem fairly few. Is this the reason for the higher heterogeneity between samples in nephrocytes compared to the other groups?
- Fig. 2A: To illustrate the purity and specificity of the samples, further genes, including at least *sns*, *Klf15*, and *kirre*, should be included.
- Bar graphs should not be shown without individual values and standard deviation.
- Volcano plots would illustrate the data for individual genes as well, which is mostly lacking.
- To become a resource for the field, the suppl. information should provide more detail, also on the level of individual genes. It would be helpful to note gene symbols for the proteomic data as well.
- Fig. 2D: Why is apoptosis elevated in nephrocytes while these cells hardly show cell death?
- Fig. 2H: Is it not surprising that pericardial nephrocytes show decreased metabolism while constantly performing endocytosis of all kinds of cargo from the hemolymph?
- Fig. 3A: Why is there so little overlap between transcriptome and proteome? This can hardly be explained by secretion alone.
- Fig. 3C: Comparing their data with the hemolymph proteome, they identify a single gene that is not part of the transcriptome. Does it not seem very unlikely that only a single gene is sequestered from the hemolymph in a cell whose entire focus is endocytosis? Is this approach valid for this question?
- Fig. 3D: Why is there so little variability in the controls?
- Fig. 4A: It would be nice to show fat body for comparison and a negative control as well. How do the authors differentiate between signal merely resulting from endocytosis at this low magnification?
- Fig. 4B-E: Why are individual values, standard deviations, and significance levels not shown?
- Fig. 4C: The brightest colors should reflect >10.0 , not <10.0 ?
- How does the function as an antimicrobial cell as proposed by the authors match with the findings by Troha et al. (PMID: 31564469), that showed that animals lacking nephrocytes are even more resistant to infection? Despite the wordy discussion, this question remains open to me. The authors describe nephrocytes as “immune-competent”, but the lack of experimental data supporting this function undermines this categorization.
- Fig. 7-8: The assumptions regarding scavenging activity should be confirmed by simple tracer studies.
- Fig. 7B-B’: Why is there no overlap between proteome and transcriptome-based KEGG?
- Does Fig. S1B not show an increase of endocytosis in GNCs contrary to what is stated in the main text (line 348)?
- Is there a difference in abundance or size of mitochondria between CNC and PNC?
- The authors should briefly discuss how their results differ from the analysis by the Perrimon lab (PMID: 35696569).
- Minor point: Page 3, line 101: „combinatorial“ should rather be „combined“

Reviewer #3

(Remarks to the Author)

The authors characterise gene expression and protein levels in two types of insect nephrocytes that represent important models for kidney research.

The team brilliantly explains the biology and results from a very complex data set which covers different developmental stages and ageing. The work is logically presented and easy to read – given the depth and breadth of the expression patterns. This is achieved through a deep understanding of the cells’ biology and intracellular processes.

This is far more than an informatics paper, it presents new insights of biological relevance that are verified and empirically tested using nephrocyte free models. There is enough information here to keep several labs going for many years!

The writing is crystal clear and I am truly struggling to find any issues – which is an ideal position to be in for a reviewer.

This is an excellent piece of work.

Abstract.

L61. Arguably ‘may’ can be removed; “This paper represents a valuable basis for...”

Intro.

L133. Is a better term ‘endoreduplication’?

Results.

Fig 3d. The bars are a little hard to differentiate – can one be filled and the other open?

Version 1:

Reviewer comments:

Reviewer #1

(Remarks to the Author)

Meyer et al. present a revised manuscript and a rebuttal letter that address all my concerns. In my opinion, the manuscript is now ready for publication.

Reviewer #2

(Remarks to the Author)

My concerns were largely addressed in the revised version of this manuscript. The authors revised relevant sections of the manuscript text and present at least some additional experimental evidence for their conclusions from the high throughput data. The revised manuscript has become stronger and now seems ready for publication.

I only have two very minor concerns:

- The controls in Fig. 3D should rather be shown for each individual value as ratio to the mean of the controls. Then it would be possible to see the variability in the controls as well.
- Please define the axes for the volcano plots.

Reviewer #1 (Remarks to the Author):

Meyer et al. characterize both types of *Drosophila* nephrocytes employing mass spectrometry based proteomics and RNA Seq based transcriptomics. While both types of nephrocytes exhibit structures similar to mammalian slit-diaphragms little is known whether garland and pericardial nephrocytes fulfil distinct functions. The authors now characterize both cell types at several time points at one, three and five weeks as adults and pericardial nephrocytes also at the 3rd instar larval stage. Functional enrichment analysis suggested that larval and adult nephrocytes, as well as garland and pericardial nephrocytes, fulfil distinct physiological functions. Pericardial nephrocytes were characterised by transcriptomic and proteomic profiles suggesting an atypical energy metabolism with very low oxidative phosphorylation rates. Moreover, the nephrocytes displayed signatures indicative of an extensive immune signalling.

Major points:

While this study characterizes differential gene and protein expression of the two different nephrocytes populations and makes it possible to formulate hypotheses about their distinct functions during the life of a fly, the data is descriptive, and changes found in gene expression were mostly not further characterized and confirmed with other methods.

In order to show the relevance of the genomics and proteomics data, it is important to focus on one pathway and delve in deeper by performing knockdown experiments analysing a potential phenotype (e.g. slit-diaphragm integrity). It would be highly interesting to dissect the functional implications of a subset of differentially expressed genes, e.g. the regulation of immune pathways in the respective nephrocytes. In my opinion showing functional implications of one of the identified targets is a prerequisite.

*Our transcriptomic / proteomic data imply that the nephrocytes are immune-competent and express a defined set of immune-response genes, especially relating to the Imd and Toll pathways. To further substantiate this indication and thus confirm the biological relevance of our datasets, we analyzed the response of the cells to bacterial infection more specifically. Expression of the antimicrobial peptide drosomycin, a major target gene of the Toll pathway, was strongly upregulated in nephrocytes of *Erwinia carotovora* (gram-negative bacteria) infected animals, relative to non-infected controls, which verifies that PNCs indeed exhibit significant immune competence. On the other hand, expression of defensin, an antimicrobial peptide active predominantly against gram-positive bacteria, was not induced, confirming specificity of the response. Corresponding data are shown in a revised Fig. 4.*

The results and the discussion sections have been amended to include these information.

Altogether, the study is interesting and novel to the field, but needs further investigation.

Minor points:

Table 1: Why were transcriptomics and proteomics performed with 3rd instar larvae PNCs but not GNCs, for a full characterization and understanding of the aging process of both cell types the analysis of 3rd instar larvae GNCs might be interesting.

Transcriptomic and proteomic analyses of 3rd instar larval GNCs were included. Data are shown in a novel Fig. S3.

Fig.2 F: Are there also differences in the proteome analysis of 1 week old GNCs compared to total fly?

Yes, there are specific differences. First and foremost, the high OXPHOS rate of the GNCs is reflected at both the transcriptome and proteome levels. Corresponding results are shown in Fig. S1.

Fig.4 B: As Mtk is the only gene that is highly expressed in GNCs an analysis of Mtk GFP-reporter in GNCs might be interesting.

We analyzed the Mtk GFP-reporter as suggested and detected distinct signals in both the GNCs and the PNCs. Results are shown in a new Fig. 4D.

In addition, we have re-evaluated our data presentation and now show it as individual "Reads Per Kilobase Million" (RPKM) values for all samples analyzed (total animals; PNCs; GNCs). Expression fold changes in PNCs and GNCs, relative to the total animals, are now shown in a new panel below the main figure (see amended Fig. 4B, D, E).

Fig.6B: Please name the respiratory chain complexes in the figure for better understanding, e.g. NADH dehydrogenase, etc.

The figure was amended accordingly.

Fig. 7B, C, B', C': Please increase the size of the labeling.

The Figure was amended accordingly.

Fig. 8: Please increase the size of the labeling for C,D,E,F,H, I

The figure was amended accordingly.

Fig.9: Please sort the columns with increasing age, e.g starting with the larval stage, for easier perusal.

The figure was amended accordingly.

Discussion: Please discuss the impact of the collagenase treatment on nephrocytes and the basement membrane and potential impact this could have had on the results.

The discussion was amended to include this issue.

Reviewer #2 (Remarks to the Author):

The manuscript by Meyer et al. delves into the distinct subtypes of *Drosophila* nephrocytes across various developmental stages through transcriptome and proteome analyses. They adopt a meticulous approach of manually sorting these cells to ensure proper sample homogeneity. In that manner, the authors lay important groundwork for this podocyte model, nicely confirming that pericardial and garland nephrocytes are clearly distinct cell types with divergent metabolism. The authors identify two candidates for the poorly understood secretory function of nephrocytes and

propose an antimicrobial role for nephrocytes. However, the data presentation should be improved. Trying to tackle many aspects, additional experimental data to support the conclusions drawn from the high throughput data remains fairly limited.

- The last section of the introduction on page 5 reads more like the first part of a results section. *The corresponding section has been largely deleted.*

- Fig. 2A: 30 cells per measurement in GCN seem fairly few. Is this the reason for the higher heterogeneity between samples in nephrocytes compared to the other groups?

In initial experiments, we found that 30 cells were sufficient for the analyses performed. Higher cell numbers had no significant influence on heterogeneity.

- Fig. 2A: To illustrate the purity and specificity of the samples, further genes, including at least *sns*, *Klf15*, and *kirre*, should be included.

The suggested genes were included into Fig. 2A.

- Bar graphs should not be shown without individual values and standard deviation.

Individual values, standard deviation, and significance analyses are now shown for all bar graphs.

- Volcano plots would illustrate the data for individual genes as well, which is mostly lacking.

Volcano plots for all comparisons are shown in a new Figure S5.

- To become a resource for the field, the suppl. information should provide more detail, also on the level of individual genes. It would be helpful to note gene symbols for the proteomic data as well.

Gene symbols were included in all proteomic data tables.

- Fig. 2D: Why is apoptosis elevated in nephrocytes while these cells hardly show cell death?

*In view of this unexpected result, we took a closer look at the significantly affected genes underlying the GO term "apoptosis". These were: *bsk*, *crc*, *eip74EF*, *jra*, *kay*, *parp*, *wts*, *traf4*, *marf*, *eIF5*, *wgn*, *egr*, *buffy*, *puc*, and *eip93F*. We found that most of the genes were also linked to other GO terms, most notably the MAPK, Hippo, and Toll and IMD signaling pathways. The fact that the induction of apoptosis is one of the many cellular functions of these signaling pathways could be the reason for the GO annotation. However, we have no indication that apoptosis actually occurs in these cells.*

These considerations were included into the results section.

- Fig. 2H: Is it not surprising that pericardial nephrocytes show decreased metabolism while constantly performing endocytosis of all kinds of cargo from the hemolymph?

This is indeed an interesting observation, which may indicate that at least part of the endocytosed cargo is stored within the cells rather than being metabolized. We included this consideration into the corresponding section of the results.

- Fig. 3A: Why is there so little overlap between transcriptome and proteome? This can hardly be explained by secretion alone.

The main reason for the seemingly small overlap between transcriptome and proteome is based on the fact that given our strict significance criteria, only relatively few proteins were identified as increased in the nephrocytes, while in the same comparison many more transcripts were found.

However, for the GNCs, 15 of the 31 identified proteins had equally altered transcript levels (48 %), while for the PNCs, 91 of the 152 identified proteins had equally altered transcripts (60 %, Fig. 3A). These numbers are in good agreement with or even exceed the proteome/transcriptome overlaps reported in similar studies from other laboratories.

- Fig. 3C: Comparing their data with the hemolymph proteome, they identify a single gene that is not part of the transcriptome. Does it not seem very unlikely that only a single gene is sequestered from the hemolymph in a cell whose entire focus is endocytosis? Is this approach valid for this question?

In this experiment, we compared our transcriptomic and proteomic datasets with published data on the composition of the haemolymph proteome to identify proteins that are sequestered by the nephrocytes. In this regard, we selected candidates that were present in the haemolymph and in our proteomic dataset, but absent in the corresponding transcriptome. Interestingly, we found that many haemolymph circulating proteins were also produced by the cells (Fig. 3C). These candidates were not included into the list of sequestered proteins. However, although they are expressed by the nephrocytes, the corresponding factors can of course also be sequestered by them. Thus, the focus on the CG6409 gene is indeed a bit misleading.

We have amended the the corresponding part of the results section accordingly.

- Fig. 3D: Why is there so little variability in the controls?

All individual gene expression values were normalized to the respective control. To enable an easy comparison, all control values were set to 1.

- Fig. 4A: It would be nice to show fat body for comparison and a negative control as well. How do the authors differentiate between signal merely resulting from endocytosis at this low magnification?

In this experiment, we used GFP-reporter lines to measure expression of the depicted genes, rather than GFP-tagged peptide constructs. Therefore, the expressed GFP is present exclusively in the cytosol of the expressing cells and not secreted by them. Endocytic processes are not affecting the readout of our experimental setup.

A negative control (w^{1118}) was included. Fat body cells are present in all preparations shown (see also the new Fig. 4F'' for comparison) and do not exhibit any signal above background.

- Fig. 4B-E: Why are individual values, standard deviations, and significance levels not shown?

Individual data points, standard deviations, significance levels, and fold changes are now shown for all analyzed genes.

- Fig. 4C: The brightest colors should reflect >10.0, not <10.0?

Yes; the figure was amended accordingly.

- How does the function as an antimicrobial cell as proposed by the authors match with the findings by Troha et al. (PMID: 31564469), that showed that animals lacking nephrocytes are even more resistant to infection? Despite the wordy discussion, this question remains open to me. The authors describe nephrocytes as “immune-competent”, but the lack of experimental data supporting this function undermines this categorization.

The findings by Troha et al showed that nephrocytes of wild-type flies removed PGN from the circulation via endocytosis and subsequent lysosomal degradation. This process was impaired in Klf15-null flies, resulting in excess PGN in haemolymph and constitutive activation of the Toll pathway. The increased resistance to infection was therefore due to the absence of the scavenger activity normally exerted by the nephrocytes.

*Our transcriptomic / proteomic data now imply that the nephrocytes themselves are immune-competent and express a defined set of immune-response genes, especially relating to the Imd and Toll pathways. To further substantiate this indication, we analyzed the response of the cells to bacterial infection more specifically. Expression of the antimicrobial peptide drosomycin, a major target gene of the Toll pathway, was strongly upregulated in nephrocytes of *Erwinia carotovora* infected animals, relative to non-infected controls, which confirms that PNCs indeed exhibit significant immune competence. Corresponding data are shown in a revised Fig. 4.*

The results and the discussion sections have been amended to include these information.

- Fig. 7-8: The assumptions regarding scavenging activity should be confirmed by simple tracer studies.

Corresponding experiments were done and confirmed a reduction in endocytic activity with age. The highest endocytosis rates were observed in 3rd instar larval cells, which decreased slightly in the 1-week-old adult cells and almost disappeared in the 5-week-old adult cells.

The results have been included in the main text and are shown in a new Fig. S4.

- Fig. 7B-B': Why is there no overlap between proteome and transcriptome-based KEGG?

Interestingly, we observed this lack of overlap not only in larval PNCs (Fig. 7B-B'), but also in larval GNCs (Fig. S3B-B'). Thus, larval nephrocytes in particular appear to be characterized by a rather low transcriptome/proteome correlation. Whether this phenomenon is due to reduced transcript or protein stability or to other molecular events such as altered translation efficiency remains unclear at this point.

The results section has been amended to include these information.

- Does Fig. S1B not show an increase of endocytosis in GNCs contrary to what is stated in the main text (line 348)?

Line 348 refers to proteome data that showed no increase in endocytosis. However, since the transcriptomic data indeed indicated an increase in endocytotic efficiency (Fig. S1B), we removed the corresponding statement from the main text.

- Is there a difference in abundance or size of mitochondria between GNC and PNC?

We analyzed mitochondrial abundance in PNCs and GNCs via mitotracker staining and found a slight, yet significant increase in mitochondria in the latter cells. Data are shown in a novel Fig. S3.

- The authors should briefly discuss how their results differ from the analysis by the Perrimon lab (PMID: 35696569).

The discussion was amended accordingly.

- Minor point: Page 3, line 101: „combinatorial“ should rather be „combined“

Correction has been made.

Reviewer #3 (Remarks to the Author):

The authors characterise gene expression and protein levels in two types of insect nephrocytes that represent important models for kidney research.

The team brilliantly explains the biology and results from a very complex data set which covers different developmental stages and ageing. The work is logically presented and easy to read – given the depth and breadth of the expression patterns. This is achieved through a deep understanding of the cells’ biology and intracellular processes.

This is far more than an informatics paper, it presents new insights of biological relevance that are verified and empirically tested using nephrocyte free models. There is enough information here to keep several labs going for many years!

The writing is crystal clear and I am truly struggling to find any issues – which is an ideal position to be in for a reviewer.

This is an excellent piece of work.

Abstract.

L61. Arguably ‘may’ can be removed; “This paper represents a valuable basis for...”

Text was amended accordingly.

Intro.

L133. Is a better term 'endoreduplication'?

Text was amended accordingly.

Results.

Fig 3d. The bars are a little hard to differentiate – can one be filled and the other open?

The suggested amendment was included.

We would like to thank all reviewers for their thorough review and constructive suggestions for improvement!

Reviewer #1 (Remarks to the Author):

Meyer et al. present a revised manuscript and a rebuttal letter that address all my concerns. In my opinion, the manuscript is now ready for publication.

Reviewer #2 (Remarks to the Author):

My concerns were largely addressed in the revised version of this manuscript. The authors revised relevant sections of the manuscript text and present at least some additional experimental evidence for their conclusions from the high throughput data. The revised manuscript has become stronger and now seems ready for publication.

I only have two very minor concerns:

- The controls in Fig. 3D should rather be shown for each individual value as ratio to the mean of the controls. Then it would be possible to see the variability in the controls as well.

The suggested amendmend was included in Fig. 3.

- Please define the axes for the volcano plots.

The suggested amendmend was included in Fig. S6.

We would like to thank all reviewers for their thorough review and constructive suggestions for improvement!